This manuscript was compiled using the Copernicus template for preprint submission to Nonlinear Processes in Geophysics.

# Long-window tandem variational data assimilation methods for chaotic climate models tested with the Lorenz 63 system

Philip David Kennedy[1], Abhirup Banerjee[1], Armin Köhl[1], and Detlef Stammer[1]

[1]Universität Hamburg, Fakultät für Mathematik, Informatik und Naturwissenschaften, Fernerkundung & Assimilation, Bundesstr. 53, 20146 Hamburg, Deutschland

**Correspondence:** Philip David Kennedy (philipdkennedy@physics.org)

**Abstract.** 4D-variational data assimilation is applied to the Lorenz '63 model to introduce a new method for parameter estimation in chaotic climate models. The approach aims to optimise an Earth system model (ESM), for which no adjoint exists, by utilising the adjoint of a different, potentially simpler ESM. This relies on the synchronisation of the model to observed
data. Dynamical state and parameter estimation (DSPE) is used to stabilise the tangent linear system by reducing all positive Lyapunov exponents to negative values, thereby improving parameter estimation by enabling long assimilation windows. The method introduces a second layer of synchronisation between the two models, with and without an adjoint, to facilitate linearisation around the trajectory of the model for which no adjoint exists. This is achieved by synchronising two Lorenz '63 systems, one with and the other without an adjoint model. Results are presented for an idealised case of identical, perfect models and
for a more realistic case in which they differ from one another. If employed on a high-resolution ESM for which a coarse resolution adjoint exists, the method will save computational resources as only one forward run with the full high-resolution ESM per iteration is needed. It is demonstrated that there is negligible error and uncertainty change compared to the traditional optimisation of a full ESM with an adjoint. Stemming from this approach, it is shown that the synchronisation between two identical models can be used to filter noisy data in a dynamical way which reduces the parametric uncertainty of the optimised
model by approximately one third. Such a precision gain could prove valuable for seasonal, annual, and decadal predictions.

## 1   Introduction

The time evolution of the Earth system can be simulated using numerical Earth system models (ESMs). Provided these models skilfully describe the system's time evolution and observed processes, they can be used to forecast future states of the system
as long as accurate initial conditions exist. Data assimilation is a powerful tool to bring ESMs into agreement with the observed climatic state by combining data with the numerical model while preserving dynamic principles governing the system (Wunsch and Heimbach, 2006; Nichols, 2010) while also attempting to further improve the ESM's predictive skills.

There are two common assimilation approaches typically used to incorporate observations into a model: *sequential* and *variational* data assimilation schemes (Wunsch, 1996). *Sequential* data assimilation (Bertino et al., 2003) involves the application

of a filter, most commonly Kalman filters (Kalman, 1960; Evensen, 1994, 2003; Tippett and Chang, 2003; Houtekamer and Mitchell, 2001). This technique merges a predicted state with observations at each analysis time step by estimating a joint probability distribution between the two by taking into account their respective modelling and observational uncertainties. Variants of the Kalman filter technique include: extended Kalman filters, ensemble Kalman filters, and square-root filters (Bar-Shalom et al., 2004; Simon, 2006; Evensen, 2003; Van Der Merwe and Wan, 2001; Tippett et al., 2003). They all share a similar basic procedure while differing in case specific variations of the methodology. The strength of all filtering techniques is that the *sequential* procedure allows for real-time assimilation of observations, for example in initialised numerical weather forecasting.

In contrast, *variational* data assimilation (Le Dimet and Talagrand, 1986) estimates a joint probability distribution over an extended period of time by minimising a scalar cost function, defined as the quadratic misfit between the model trajectory and all available observations within a defined time window. The most common approaches include four-dimensional *variational* assimilation (4D-var.) (Rabier and Liu, 2003), three-dimensional *variational* data assimilation (3D-var.) (Gustafsson et al., 2001), weak and strong constraint 4D-var (Tremolet, 2006; Fisher et al., 2011), and ensemble *variational* filters including 4DEnVar (Desroziers et al., 2014). *Variational* data assimilation is a useful technique for solving both initial value and parameter estimation problems (Evensen et al., 2022; Goodliff et al., 2015; Ruiz et al., 2013; Zou et al., 1992). It will be exclusively used in this study.

The 4D-var. approach utilises an adjoint of the model to iteratively minimise the model-data misfit by adjusting control variables (Tett et al., 2017; Lyu et al., 2018; Köhl and Willebrand, 2002; Allaire, 2015; Navon, 2009). The adjoint equations of a fully non-linear model are derived from the forward equations using integration by parts. In the data assimilation context this can be used to numerically calculate the gradient of the cost function which is subsequently used to find the cost functions minimum in an iterative procedure. Adjoint models have also been widely used for sensitivity analysis in meteorology and oceanography (Hall et al., 1982; Hall and Cacuci, 1983; Hall, 1986; Marotzke et al., 1999; Stammer et al., 2016); this includes calculating sensitivity with respect to lateral boundary conditions (Gustafsson et al., 1998), estimating the sensitivity of the 2m surface temperature with respect to the sea surface temperature, sea ice, and sea surface salinity (Stammer et al., 2018). In practice, the primary limitation in finding the minimum of the cost function is the large amount of computational resources required due to non-linear or chaotic elements of the system.

In the context of a full non-linear ESM, the use of adjoint models faces several challenges. Applying an adjoint model to a state-of-the-art Earth system problem is primarily limited by the very large number of state variables $\mathcal{O}(10^7 - 10^8)$ which requires significant computational resources and observational constraints. However, more fundamental is the fact that non-linear dynamics of the system limit the applicability of adjoint methods to the Earth system predictability time scale. This can lead to exponentially growing adjoint sensitivities as a result of multiple local minima in the cost-function. Under such circumstances spikes occur in the estimated gradients and the cost function becomes very rough by showing an increasing number of local minima (Köhl and Willebrand, 2002; Lea et al., 2000). Fortunately, the problem can be mitigated through synchronisation which removes the non-linear or chaotic dynamics leading to a smooth cost function (Abarbanel et al., 2010).

This method allows for the extension of the assimilation window beyond the predictability time-scale, provided that sufficient observations are available. However, this solution comes at the expense of a violation of the original model equations.

The creation of an adjoint model code from the forward code usually requires considerable effort. Automatic differentiation tools, such as Giering and Kaminski (1998); Hascoet and Pascual (2013) were developed to aid in this step. But substantial changes to the forward model code are required unless it was already developed with the adjoint modelling in mind. Stammer et al. (2018) created the first adjoint of an intermediate complexity fully coupled earth system model that is automatically created from the forward model by automatic differentiation using the TAF compiler, called the Centrum für Erdsystemforschung und Nachhaltigkeit (CEN) Earth System Assimilation Model (CESAM). The adjoint of this intermediate-complexity model is intended to be utilised for tuning more complex CMIP-type models through parameter estimation since the basic underlying physics is very similar. Otherwise this is a manual process with considerable ambiguity in the choice of parameters (Mauritsen et al., 2012).

Therefore, we propose a novel framework in which we use two climate models both coupled through synchronisation, one with a high complexity and the other of intermediate complexity for which an adjoint exists to address the second problem. The technique also has a much wider range of additional applications, since resolutions using the adjoint method lag behind those applications featuring simpler assimilation methods as variational methods are typically a factor of 100 more costly than running the associated forward model. For example, the global GECCO3 ocean reanalysis based on the adjoint method (Köhl, 2020) features only a nominal resolution of $0.4°$, while for instance the GOFS 3.1 (Laboratory, 2016) based on 3D-Var (Cummings and Smedstad, 2013) features $1/12°$ resolution. Employing coarser versions of the adjoint while still running the forward model with full resolution could significantly reduce the cost of the assimilation effort. Therefore, the objective of this paper is to investigate the accuracy and precision of such a synchronised data assimilation approach. We perform this test using a Lorenz '63 model.

The Lorenz '63 system (Lorenz, 1963) is a well-established proxy model of chaotic fluid systems, such as the atmosphere (Gauthier, 1992; Miller et al., 1994; Pires et al., 1996; Stensrud and Bao, 1992; Kravtsov and Tsonis, 2021; Huai et al., 2017; Yang et al., 2006; Daron and Stainforth, 2015; Errico, 1997). The advantage is that it can be used to rapidly evaluate parameter estimation techniques in data assimilation schemes prior to their application in a full ESM with low computational resource requirements. New modelling techniques can thus be trialled in fast experiments (Pasini and Pelino, 2005; Tandeo et al., 2015; Goodliff et al., 2020; Marzban, 2013; Yin et al., 2014). It can also be used in a wide range of other applications including, but not limited to, data assimilation, stochastic modelling terms, and predictions (Du and Shiue, 2021; Cameron and Yang, 2019; Pelino and Maimone, 2007). The system generates a three-dimensional, time-varying trajectory which with variation of both model parameters and/or initial conditions will produce very different trajectories. Thus, it is an ideal test bed for non-linear modelling in a number of fields (Hirsch et al., 2013). The Lyapunov exponent of the Lorenz '63 model is directly dependent upon its parameters making it ideal for climatological parameter estimation experiments. For our specific case these properties make it ideal to evaluate our techniques merits. In a previous study, Lyu et al. (2018) used the Lorenz '63 model and its adjoint to fit a single parameter $\rho$ and the initial conditions $(x, y, z)$ to observations. This present study builds on

Lyu et al. (2018) to simultaneously fit all three model parameters and use a model with an adjoint to optimise the parameters of another model without one.

The structure of the remaining paper is as follows: In Section 2 we outline the methodology of synchronisation, the cost function and the adjoint method. Section 3 introduces the Lorenz '63 model, describes our reference setup, before introducing our two novel multi-model methods, describing our minimisation algorithm, and detailing our statistical metrics for evaluating results. Section 4 shows and discusses the results of our multi-model setups, using a single model setup as a baseline for comparison. The results of introducing a mismodelling term to the adjoint model are also included. A summary and concluding remarks are given Section 5.

## 2 Methodology

### 2.1 Synchronisation

In chaotic systems, integrating over periods longer than the predictability time scale creates problems for accurate parameter estimation. This is due to exponentially growing gradients, and a maximum likelihood estimate with an increasing number of local maxima (Köhl and Willebrand, 2002; Lea et al., 2000). The non-linear or chaotic dynamics, which detrimentally effect the maximum likelihood estimate, can be removed by synchronisation (Abarbanel et al., 2010; Sugiura et al., 2014) which transforms the chaotic model into one with linear dynamics without positive Lyapunov exponents leading to maximum likelihood functions with one unique maxima. This can be implemented into a generic model of ordinary differential equations,

$$\dot{\boldsymbol{x}}(t) = f(\boldsymbol{x}(t), \boldsymbol{\theta}, t) \tag{1}$$

where $\boldsymbol{x}(t)$ is the state vector, $\boldsymbol{\theta}$ is the parameter vector, and $t$ is the time, synchronisation can be incorporated by adding a term which penalises the difference between the model and observations. This term is simply added to the equations

$$\dot{\boldsymbol{x}}(t) = f(\boldsymbol{x}(t), \boldsymbol{\theta}, t) + \alpha(\boldsymbol{x}_o(t) - \boldsymbol{x}(t)) \tag{2}$$

where $\alpha$ is the synchronisation coefficient and $\boldsymbol{x}_o(t)$ is the observation state vector.

According to the law of large numbers both with perfect models and in the presence of noise, the precision of the recovered parameters will improve with increasing window length since more data is integrated into the estimation. Similar benefits could be achieved by averaging estimates obtained over short windows, for which no synchronisation is necessary. However, underlying restrictions differ. For synchronisation, noise affects the state over the entire window, whereas for short windows noise effects are transported. Short window assimilation can be of benefit in perfect model settings from the error growth as suggested by the quasi-static variational assimilation (QSVA) framework (Pires et al., 1996) due to fact that sensitivities increase exponentially with time in chaotic models. The analogue of this QSVA effect in the Dynamical State and Parameter Estimation (DSPE) method (Abarbanel et al., 2009) is the attempt to reduce the synchronisation parameter as the optimisation progresses and parameters move closer to their true values. Since errors and sensitivities grow exponentially, feasible window

lengths in QSVA have a maximum value due to limited numerical precision. Similarly, synchronisation parameters cannot
approach zero for assimilation windows much larger than the predictability limit, because synchronisation will eventually fail
if positive Lyapunov exponents exist (Quinn et al., 2009). We note that the reasoning for the need of long assimilation windows
is somewhat different in the context of full ESM, for which it is essential to resolve long time scale physical mechanisms
impacted by the specific choice of parameters, such as air-sea interactions of advection time scales in the ocean.

## 2.2 The cost function

As previously mentioned, in the context of variational data assimilation a cost function, $J$, must be introduced which measures
the quadratic misfit between the model trajectory and observations. For the case of perfectly known initial conditions, but
uncertain parameters $\boldsymbol{\theta}$, $J$ takes the generic form

$$J = \frac{1}{2N} \left(\boldsymbol{\theta} - \boldsymbol{\theta}_b\right)^T \frac{1}{\sigma_{\boldsymbol{\theta}_b}^2} \left(\boldsymbol{\theta} - \boldsymbol{\theta}_b\right) + \frac{1}{2N} \int_0^N dt \left(\boldsymbol{x}_o(t) - \boldsymbol{h}(\boldsymbol{x}(t))\right)^T \frac{1}{\sigma_{\boldsymbol{x}_o}^2} \left(\boldsymbol{x}_o(t) - \boldsymbol{h}(\boldsymbol{x}(t))\right) \tag{3}$$

where $N$ is the total integration time, $\sigma_{\boldsymbol{x}_o}$ is the known uncertainty associated with the observational noise, $\boldsymbol{h}(\boldsymbol{x}(t))$ is the mea-
surement function on model's predicted state vector $\boldsymbol{x}$, and $\boldsymbol{x}_o$ is the observation state vector. The prior parameter information
with the associated uncertainty is denoted by $\boldsymbol{\theta}_o$ and $\sigma_{\boldsymbol{\theta}_b}$, respectively. The global minimum of this function is the maximum
likelihood estimate of the model parameter values relative to the observations and prior information.

## 2.3 The adjoint method and the cost function gradient

To aid in the minimisation of the cost function, it is standard practice to calculate its gradient and use this to iteratively adjust
control parameters. The adjoint model is introduced to provide these cost function gradients and requires the generation of
the adjoint of the forward model equations. The resulting adjoint model can then be integrated in the reverse direction to give
the gradient of the cost function. The background term in Eq. 3 can be omitted assuming a well-posed problem, without prior
information on the parameter. Therefore, the gradient of the cost function with respect to the parameters is

$$\boldsymbol{\nabla}_{\boldsymbol{\theta}} J = -\frac{1}{N} \int_0^N dt \boldsymbol{\lambda}(t) \partial_{\boldsymbol{\theta}} f(\boldsymbol{x}(t), \boldsymbol{\theta}, t), \tag{4}$$

where $N$ is again the total integration time period, $\boldsymbol{\lambda}(t)$ is the adjoint vector at time $t$, and $\partial_{\boldsymbol{\theta}} f(\boldsymbol{x}(t), \boldsymbol{\theta}, t)$ is the partial
differential of the model with respect to the model parameters at time $t$.

## 3 Experimental setup

### 3.1 Lorenz '63 model

In this study, we use the Lorenz '63 system for all our experiments (Lorenz, 1963). The model is defined by the equations:

$$\frac{dx}{dt} = \sigma(y - x), \tag{5a}$$

$$\frac{dy}{dt} = \rho x - y - xz, \tag{5b}$$

$$\frac{dz}{dt} = xy - \beta z \tag{5c}$$

where $\boldsymbol{x} = (x, y, z)$ are the state variables at each given time step and $\boldsymbol{\theta} = (\sigma, \rho, \beta)$ are the model parameters. Throughout this article, we integrate all our models using the fourth-order Runge-Kutta method with a step size of $\Delta t = 0.01$ and total time period of 100 time units [TUs]. This system of equations will be subsequently referred to as the true model with the parameters $\boldsymbol{\theta}_t = (10, 28, 8/3)$. This true model is used to generate pseudo-observation which will be used for synchronisation, data assimilation, and parameter estimation. Noise is included in these pseudo-observations by adding random values from a Gaussian distribution centred at zero relative to the true trajectory. The random noise magnitudes are bounded to 25% of the Lorenz '63 system's standard deviation. These pseudo-observations will be labelled as $\boldsymbol{x}_o = (x_o, y_o, z_o)$.

### 3.2 Reference setup (single model)

To quantify the efficacy of our novel method we outline a reference setup for a synchronised Lorenz '63 framework similar to those described in Yang et al. (2006); Lyu et al. (2018). We expand the Lorenz '63 model (Eq. 5) by adding synchronisation terms which then reads:

$$\frac{dx}{dt} = \sigma(y - x) + \alpha(x_o - x), \tag{6a}$$

$$\frac{dy}{dt} = \rho x - y - xz + \alpha(y_o - y), \tag{6b}$$

$$\frac{dz}{dt} = xy - \beta z + \alpha(z_o - z). \tag{6c}$$

Here $\alpha$ is the synchronisation coefficient and $\boldsymbol{x}_o = (x_o, y_o, z_o)$ are the pseudo-observations generated from the true model. We set the initial parameter values of this model at the start of the optimisation as the true system values plus a 10%-error. This gives $\sigma_a = 11$, $\rho_a = 30.8$, and $\beta_a = 44/15$ which will act as our initial values for the parametric fit. The initial conditions will remain unchanged compared to the true model as our interests are exclusively in climatic parameter estimation. Synchronisation will occur at every time step in all our setups and its coefficient will also be present in the adjoint equations. The significance of this will be discussed in Section 4, as $\alpha$ has a critical role in the precision and accuracy to which parameters can be estimated, due to its influence on both the cost function and its gradient.

For each state variable in Eq. 6 a synchronisation term is included. There are seven possible combinations of these state variables which can be synchronised. The effect of each of the possible choices on the root mean squared error (RMSE) between

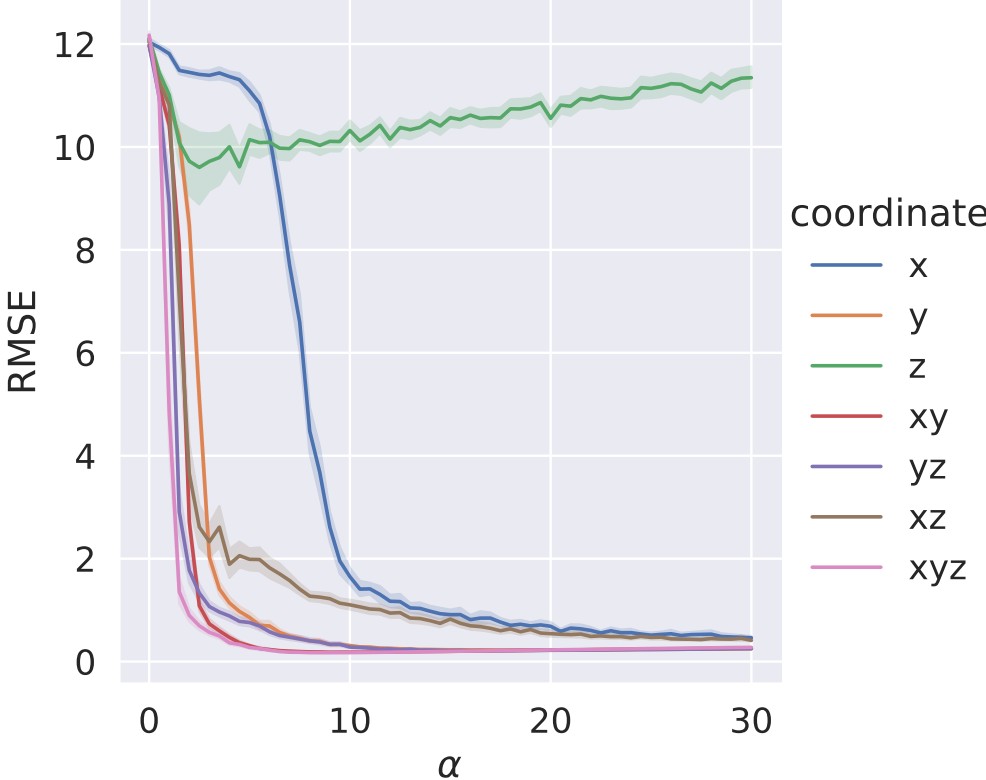

**Figure 1.** The RMSE between the true and reference model trajectories in seven different synchronisation scenarios. The synchronisation constant $\alpha$ is varied from 0 to 30 in steps of 0.5. The solid line is the median and the shaded area is the 68% percentile interval for the ensemble of 100 experiments carried out.

the true and adjoint systems by varying the synchronisation constant $\alpha$ from 0 to 30 is shown in Fig. 1. Noise was added (with zero mean and $\sqrt{2}$ standard deviation) to the true when constructing the pseudo-observations. The figure demonstrates that synchronising the $z$-component is ineffective at reducing the RMSE (Yang et al., 2006). In contrast, synchronising both $x$ and $y$ prove effective, with $y$ leading the lowest RMSE values of the single variable for all values of $\alpha$. Synchronising $xyz$ and $xy$
achieve the most effective reduction in RMSE for the lowest value of $\alpha$. It can be seen in the figure that synchronising $z$ can lead to model instability. Thus, we choose to only synchronise $x$ and $y$ in the following research to achieve more stable and accurate results with negligible precision loss.

The Lorenz '63 attractors for the trajectories of the true model and that with an adjoint are shown in Fig. 2a without synchronisation. A large divergence is visible between the trajectories. However, if synchronisation is introduced the trajectories
become very similar, as shown in Fig. 2b. There is now significant overlap between their kernel density estimations (KDEs). KDEs represent a smoothed estimate of the PDF for the model trajectory over a given time period. This allows for convenient

visual comparison of trajectories. A more numerically rigorous method to check for effective synchronisation will be discussed in Section 4.

For all subsequent experiments with our setups, a parallel experiment will be performed with this reference setup. The differences in the results can then be compared to evaluate the advantages and disadvantages of the novel techniques.

## 3.3 Multi-model data assimilation

A multi-model tandem technique is now considered, which consecutively synchronises two forward models before running the adjoint of the second model backward in time. For this purpose, Eq. 6 must be modified to incorporate a consecutive synchronisation. A schematic of this setup is provided in Fig.3 and the implications of the two possible ways to calculate the cost function are discussed in the subsequent subsections.

The first model has no adjoint equations and is the target model for which we wish to optimise the parameters. The equations of model 1, which is run only in forward mode, are

$$\frac{dx_f}{dt} = \sigma(y_f - x_f) + \alpha(x_o - x_f), \tag{7a}$$

$$\frac{dy_f}{dt} = \rho x_f - y_f - x_f z_f + \alpha(y_o - y_f), \tag{7b}$$

$$\frac{dz_f}{dt} = x_f y_f - \beta z_f \tag{7c}$$

where the sub-script $f$ denotes the forward run of model 1 and sub-script $o$ denotes observations generated from true model. The system of equations for the model 2 which has an adjoint will now be modified to synchronise with the forward-only model and not the observations:

$$\frac{dx_a}{dt} = \sigma(y_a - x_a) + \alpha(x_f - x_a), \tag{8a}$$

$$\frac{dy_a}{dt} = \rho x_a - y_a - x_a z_a + \alpha(y_f - y_a), \tag{8b}$$

$$\frac{dz_a}{dt} = x_a y_a - \beta z_a \tag{8c}$$

where the sub-script $a$ denotes model 2 which has an adjoint. This model synchronises with the model 1 but never directly with the observations.

### 3.3.1 Setup 1 - state filtered data assimilation (SFDA)

Assuming that both model 1 and 2 can be thought of as representing two identical climate models, the cost function can be placed on the model 2. This allows model 1 to filter out some of the background noise on the observations before they are given to the cost function attached to model 2. Such a filtering setup would theoretically reduce parametric uncertainty below that of traditional single model data assimilation because model 1 should act to reduce the amount of noise synchronised into model 2. We will subsequently refer to this setup as state filtered data assimilation (SFDA.)

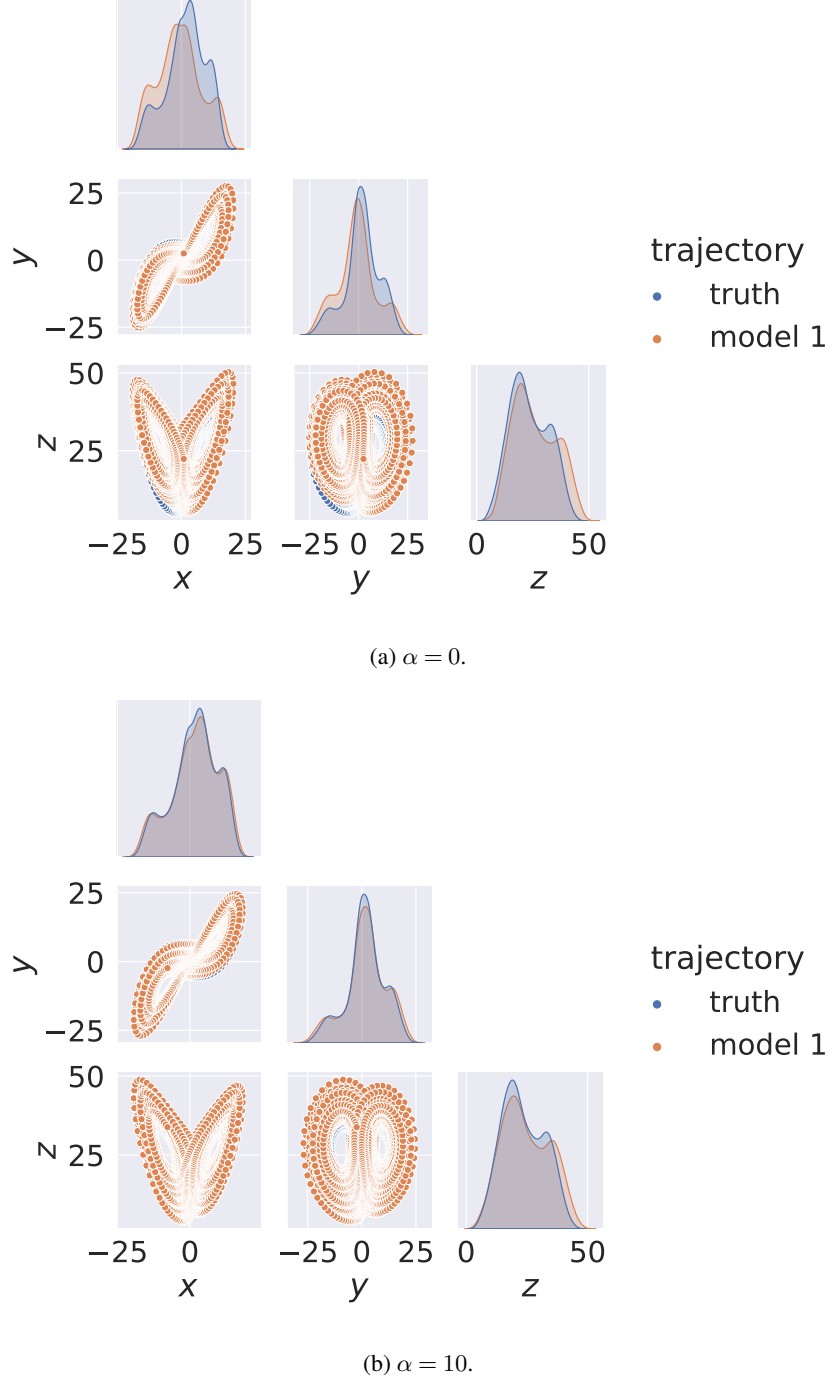

(a) $\alpha = 0$.

(b) $\alpha = 10$.

**Figure 2.** The bottom left quadrants show the Lorenz '63 true and model attractors from the main three variable orientations. The diagonal plots show kernel density estimations (KDEs). Fig. 2a shows the trajectories without synchronisation. Fig. 2b shows the trajectories with synchronisation.

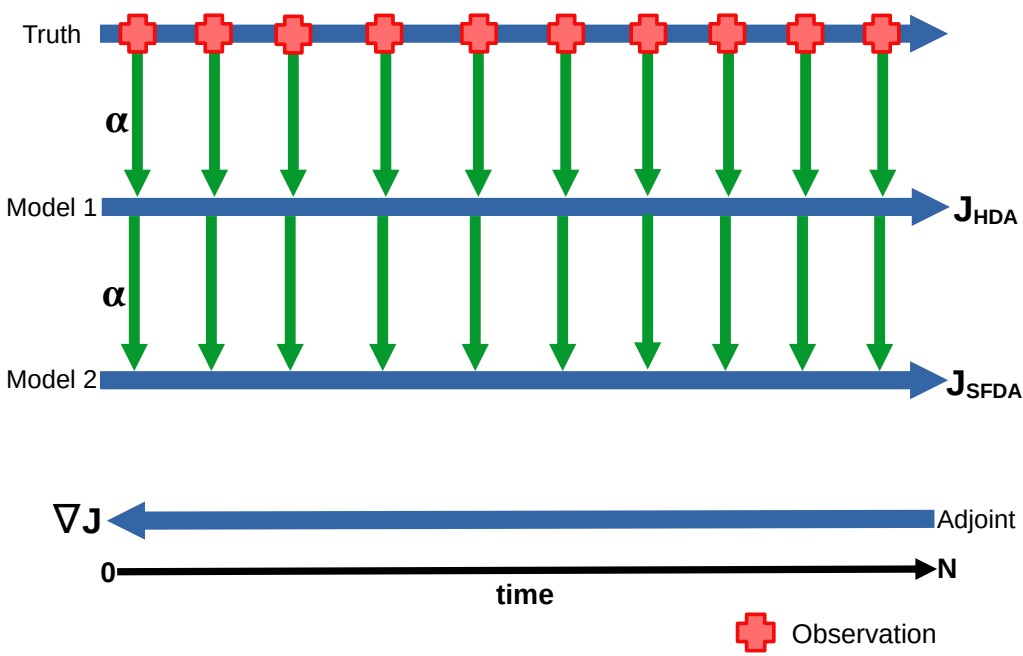

**Figure 3.** Illustration of the multi-model setup where each pseudo-observation generated from the true model includes random additive Gaussian noise. The cost function can measure the difference between the observations and either model 1 or 2 depending on the assumptions made. Both options are discussed in the text.

In SFDA the cost function acts to constrain model 2. The cost function is

$$J_{\text{SFDA}} = \frac{1}{2N} \int\limits_{t=0}^{N} dt \, (\boldsymbol{x}_o(t) - \boldsymbol{x}_a(t))^T \frac{1}{\sigma_{\boldsymbol{x}_o}^2} (\boldsymbol{x}_o(t) - \boldsymbol{x}_a(t)) \tag{9}$$

$N$ is again the total number of time steps of the assimilation window, and $\sigma_{\boldsymbol{x}_o}$ is the uncertainty associated with the observation noise. The adjoint matrix includes terms arising from the second tandem layer of synchronisation for model 2. This is given by:

$$M_{\text{SFDA}}^* = \begin{pmatrix} -(\sigma+\alpha) & \rho - z_a & y_a & 0 & 0 & 0 \\ \sigma & -(1+\alpha) & x_a & 0 & 0 & 0 \\ 0 & -x_a & -\beta & 0 & 0 & 0 \\ \alpha & 0 & 0 & -(\sigma+\alpha) & \rho - z_f & y_f \\ 0 & \alpha & 0 & \sigma & -(1+\alpha) & x_f \\ 0 & 0 & 0 & 0 & -x_f & -\beta \end{pmatrix} \tag{10}$$

which in practice is numerically evaluated using the automatic differentiation (AD) package `jax` to calculate the vector-Jacobian product.

The adjoint equation for SFDA is given by:

$$\dot{\boldsymbol{\lambda}}_{\text{SFDA}}(t) = \frac{1}{\sigma_{\boldsymbol{x}_o}^2} \left( (\boldsymbol{x}_o(t), 0, 0, 0) - (\boldsymbol{x}_a(t), 0, 0, 0) \right) \tag{11a}$$

$$- \boldsymbol{M}_{\text{SFDA}}^*(t) \boldsymbol{\lambda}_{\text{SFDA}}(t) \text{ for } t = N, ..., 0$$

with $\boldsymbol{\lambda}_{\text{SFDA}}(N) = \boldsymbol{0}.$ (11b)

These equations were derived using the method detailed in Talagrand (2010). The gradient can then be calculated with respect to the parameters $(\sigma, \rho, \beta)$ notated by the subscript $\boldsymbol{\theta}$. This yields

$$\boldsymbol{\nabla}_{\boldsymbol{\theta}} J_{\text{SFDA}} = \frac{1}{N} \int\limits_{t=N}^{0} dt \, \boldsymbol{\lambda}_{\text{SFDA}}(t) \begin{pmatrix} y_a(t) - x_a(t) \\ x_a(t) \\ -z_a(t) \\ y_f(t) - x_f(t) \\ x_f(t) \\ -z_f(t) \end{pmatrix} \tag{12}$$

which is a component-wise multiplication at each time step.

### 3.3.2 Setup 2 - Tandem data assimilation (TDA)

In this section, we want to explore if using an existing adjoint from one model could be utilised to optimise a different target model without adjoint. This will be referred to as tandem data assimilation (TDA). In TDA we assume that both models may differ in resolution or numerical formulation but are governed by the same continuum dynamics. Instead of interpolating or transforming the original model variables onto the adjoint model grid, formulation of the adjoint model through synchronisation would provide a simpler means to do this as only essential parameters need to be interpolated. Auxiliary variables and parameters will be generated by the synchronised model, including those that may not exist in the target model.

The cost function of TDA is

$$J_{\text{TDA}} = \frac{1}{2N} \int\limits_{t=0}^{N} dt \, (\boldsymbol{x}_o(t) - \boldsymbol{x}_f(t))^T \frac{1}{\sigma_{\boldsymbol{x}_o}^2} (\boldsymbol{x}_o(t) - \boldsymbol{x}_f(t)). \tag{13}$$

$N$ is the total number of time steps of the assimilation window and $\sigma_{\boldsymbol{x}_o}$ is the uncertainty associated with the observation noise. This measures the quadratic misfit between the forward-only model 1 and the observations. Model 1, $\boldsymbol{x}_f$, will be constrained by this cost function and its gradient will be calculated using the adjoint of model 2, $\boldsymbol{x}_a$. In this formulation, the two systems are no longer considered as a single synchronised one but as two separate models, one for the calculation of the trajectory and the other for calculating the gradient from its adjoint. The algorithm is also no longer exact as we only assume that model 2 ajoint will provide a good approximation as long as its trajectory follows model 1 closely and is driven by the model-data

differences of model 1. The adjoint matrix is

$$
\boldsymbol{M}^*_{\text{TDA}} = \begin{pmatrix} -(\sigma + \alpha) & \rho - z_a & y_a \\ \sigma & -(1 + \alpha) & x_a \\ 0 & -x_a & -\beta \end{pmatrix}. \tag{14}
$$

which is numerically evaluated using AD. The synchronisation with model 1 ensures that the trajectory of model 2 $x_a$ closely follows that of model 1.

The adjoint equation for TDA is given by:

$$
\dot{\boldsymbol{\lambda}}_{\text{TDA}}(t) = \frac{1}{\sigma^2_{\boldsymbol{x}_o}} (\boldsymbol{x}_o(t) - \boldsymbol{x}_f(t)) \tag{15a}
$$
$$
- \boldsymbol{M}^*_{\text{TDA}}(t) \boldsymbol{\lambda}_{\text{TDA}}(t) \text{ for } t = N, ..., 0
$$

with $\boldsymbol{\lambda}_{\text{TDA}}(N) = \boldsymbol{0}$. $\tag{15b}$

The gradient with respect to the parameters $\boldsymbol{\theta} = (\sigma, \rho, \beta)$ is calculated to be:

$$
\boldsymbol{\nabla}_{\boldsymbol{\theta}} J_{\text{TDA}} = \frac{1}{N} \int_{t=N}^{0} dt \boldsymbol{\lambda}_{\text{TDA}}(t) \begin{pmatrix} y_a(t) - x_a(t) \\ x_a(t) \\ -z_a(t) \end{pmatrix} \tag{16}
$$

which is again a component-wise multiplication at each time step. For TDA and SFDA, the trajectories of both models and adjoint vectors are stored for evaluation of the gradient.

## 3.4 Minimisation algorithm

To assimilate the data, we fit one of the synchronised models to the observations by optimising the model parameters. A cost function is constructed to calculate the misfit between observations and the model of interest. The gradient of the cost function, with respect to the model parameters, is always calculated using the adjoint method. However, the form of the adjoint will vary between the two methods we presented in Eqs. 11 and 15. The adjoint model is numerically evaluated by AD of model 2. This is done in the python package `JAX` which numerically evaluates the vector Jacobian product of the model with respect to its state variable vector (Bradbury et al., 2018). This is then integrated using an inverse Runge-Kutta scheme. Our code stores the state variables and adjoint vectors at each time step. It is also possible to carry out the entire integration using `JAX`. The process of synchronising all models, calculating the cost function and its gradient, and then adjusting model parameters is carried out iteratively by our chosen minimisation algorithm. Throughout all steps the parameter value of forward-only and adjoint models are identical and optimised simultaneously.

## 3.5 Statistical metrics

To get a more robust quantification of our setup's behaviour, it is necessary to repeat our study over a number data sets to calculate medians and percentile intervals (PIs). This allows us to examine general traits of our model without an individual

noise event obscuring trends and features of significance. Here this is done by generating 100 pseudo-data sets and assimilating each set independently. The plotting package is then directly applied to these 100 outputs to plot the median and 68% PIs. The PIs are included to illustrate the statistical spread of the results and reproducibility, not to explicitly indicate uncertainty. Hence, we choose 68% for our PIs to give a concise visualisation of the central $1\sigma$ of results. The mean percentage error and uncertainty are plotted separately to allow for quantification of both the accuracy and precision of our results. These are calculated by:

$$\text{mean \%-error} = 100\% \cdot \sqrt{\frac{1}{3} \cdot \left[ \left( \frac{\sigma - \sigma_t}{\sigma_t} \right)^2 + \left( \frac{\rho - \rho_t}{\rho_t} \right)^2 + \left( \frac{\beta - \beta_t}{\beta_t} \right)^2 \right]} \tag{17}$$

and

$$\text{mean \%-uncertainty} = 100\% \cdot \sqrt{\frac{1}{3} \cdot \left[ \left( \frac{\Delta\sigma}{\sigma_t} \right)^2 + \left( \frac{\Delta\rho}{\rho_t} \right)^2 + \left( \frac{\Delta\beta}{\beta_t} \right)^2 \right]}. \tag{18}$$

The error is calculated by percentage difference between the fitted and true parameter values. The parametric uncertainty is calculated by the minimisation algorithm using a Hessian estimate.

After parameter estimation, the optimised parameters are used to initialise a free unsynchronised run of the model. The attractors are plotted against the attractor of the true model. In all cases with synchronisation greater than or equal to the optimum value, the attractors' KDE shows precise and consistent agreement with that of the true model. Results are not displayed as there is no differences which are merit discussion. Thus, our focus in the subsequent results is to compare how the examined setups differ in terms of accuracy and precision of optimised parameters recovered.

## 4 Results

Throughout the following section we will use the single model described in Lyu et al. (2018) as our benchmark to compare the new setups against. To understand the behaviour of the setups at different operating extremes, assimilations are carried out for variations of observational noise and $\alpha$. This will help establish the optimal synchronisation strength dependent on the noise amplitude. We will also be able to compare the errors and uncertainties of the single model with our multi-model setups.

### 4.1 SFDA (setup 1)

The results from a scan of $\alpha$ are shown in Fig. 4. The single model scan has two main regions. The first, for $\alpha \leq 7.5$, is where the system is poorly synchronised leading to an inaccurate fit of the parameters and an unstable median value. The second, where $\alpha > 7.5$, is where the system is fully synchronised and recovers the true model parameters very effectively. SFDA has a higher onset of effective synchronisation than the single model setup, beginning at $\alpha = 11$. Above $\alpha = 12.5$, SFDA has consistently more accurate parameter recovery than the single model setup, while the opposite holds below $\alpha = 12.5$. The minimum error at the respective optimal $\alpha$ values are nearly identical.

Fig. 5 shows the results of two fits carried out for data with an applied noise of 25% relative to the systems' standard deviation. The mean percentage uncertainty over the three parameters is plotted for both setups. Noticing in particular the

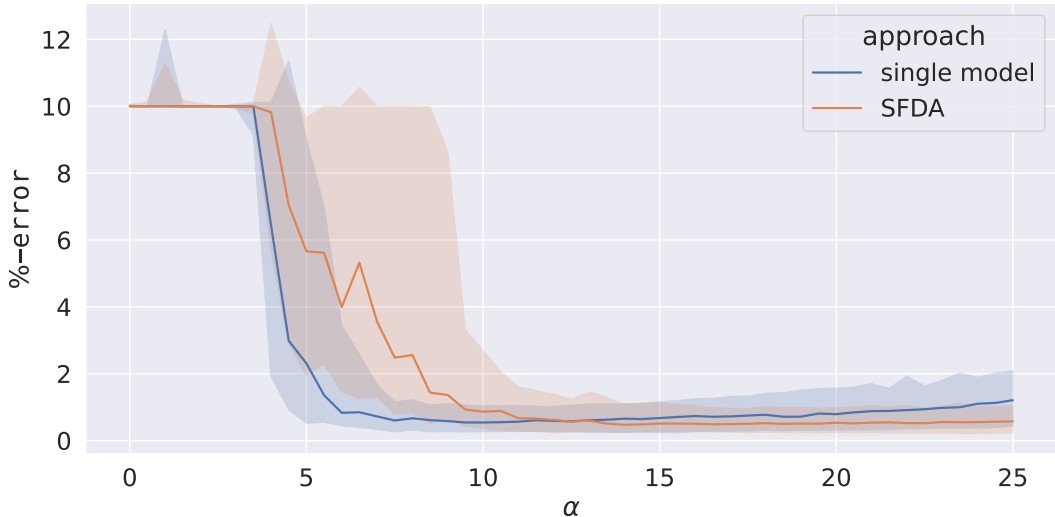

**Figure 4.** The percentage error between the true values of $(\sigma, \rho, \beta)$ and the fitted value from SFDA. A single model assimilation is included for comparison. An ensemble of 100 assimilations is carried out over 100 different data sets. The median (lines) and $68\%$ percentile intervals (shaded areas) are plotted. The noise level is $25\%$.

spread of the percentile intervals, the single model setup is found to be synchronised and have a high precision from $\alpha = 7.5$ and SFDA from $\alpha = 11.5$. Once the SFDA setup is synchronised, it is found to have a reduced uncertainty compared with the single model. SFDA is found to be approximately one third more precise than the single model setup for all value of $\alpha$ investigated. However, since SFDA requires larger $\alpha$, uncertainty values for the same $\alpha$ cannot be directly compared.

Fig. 4 suggest that the SFDA technique is more accurate than a standard single model setup at higher values of $\alpha$. Fig. 5 additionally shows that SFDA is more precise than a single model for all values of $\alpha$ after the onset of effective synchronisation. In cases where accuracy and precision are desirable, and computational resources and time are available this would advocate for the use of SFDA. However, Fig. 4 suggests that the error estimates do not represent the actual achieved accuracy of the parameter estimation. Both metrics suggest that the accuracy is less sensitive to the choice of $\alpha$ for SFDA. It is also important
to note that in the context where precision is the priority, the lowest value of $\alpha$ after effective synchronisation should be chosen. Increasing values of $\alpha$ increase the parametric uncertainty because of the associated declining sensitivity of the trajectory to parameter changes. This is also the reason why for the same $\alpha$ SFDA shows consistently better performance: the less efficient indirect constraint to the observations makes it more sensitive to the parameters.

Fig. 6 shows the results of varying the noise levels on the fitted parameter values. For all applied noise levels the quality
of the fit can be considered good as the median of the mean percentage uncertainty on the parameters remains below 0.5% even with noise levels of up to 50%. SFDA is found to have a mean error performance similar to the single model system across the range of noise levels tested. However, the spread of the error is slightly improved in the double model setup at low noise due to the forward-only model smoothing outlying observation better than a single model setup. The parametric

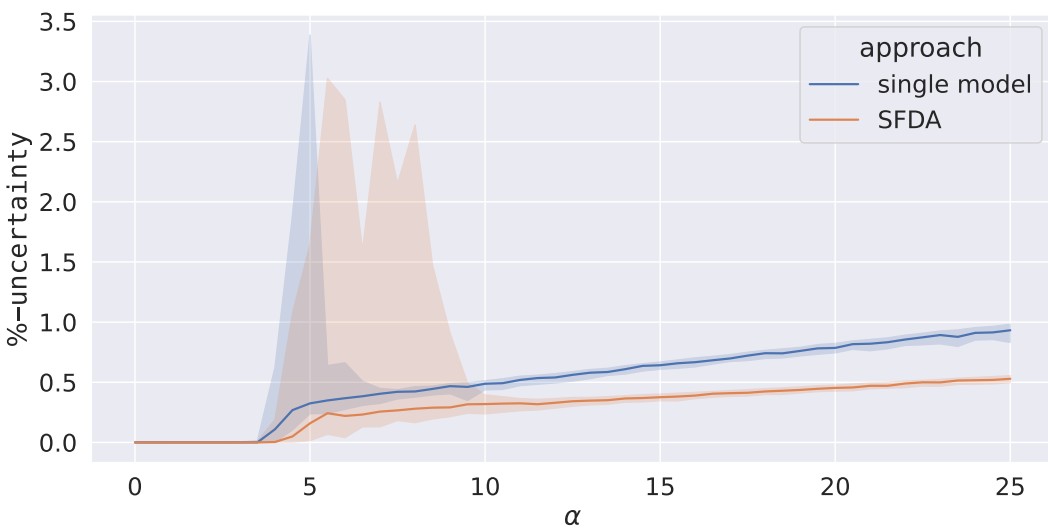

**Figure 5.** The average percentage uncertainty on the three parameters $(\sigma, \rho, \beta)$ after SFDA from the minimisation algorithm for different values of $\alpha$. A single model assimilation is included for comparison. An ensemble of 100 assimilations is carried out over 100 different data sets. The median (line) and $68\%$ percentile intervals (shaded areas) are plotted.

uncertainty is found to be consistently reduced in the double model system for all noise levels. This demonstrates the precision improvement achieved by running the forward model twice to smooth the observations before carrying out data assimilation. The consequences of this are that for smaller models, where computational resources are available and improved precision or accuracy are desirable, SFDA can reduce error and particularly decrease uncertainty.

## 4.2  TDA (setup 2)

Similar to SFDA, TDA results are evaluated in terms of percentage error and uncertainty estimates against the single model with a data noise level of $25\%$. Error estimates are shown in Fig. 7 as function of $\alpha$. For $\alpha \geq 3$ synchronisation starts to set in and parameter estimation begins to improve. The system is only synchronising effectively for $\alpha \geq 7.5$ to consistently recover the true model parameters, as is visible from the small spread of the error. The TDA scan follows the behaviour of the primary model very closely with no visible disadvantage.

The mean percentage uncertainty over the three parameters is plotted in Fig. 8 for both setups. TDA is found to have almost identical uncertainty to the single model. The plot consists of two regions. The first, for $\alpha < 7.5$, is the region where the model is not yet consistently synchronised producing high variability depending on the specific noise. The second, for $\alpha \geq 7.5$, is where the system is consistently synchronised. The minimum median of the mean parametric uncertainty, after consistent synchronisation begins, is $\approx 0.35\%$ and achieved at $\alpha = 7.5$. The subsequent increase in uncertainty is due to the reduced the parametric sensitivity associated with the increased $\alpha$, thereby reducing the curvature of the cost function at the minimum.

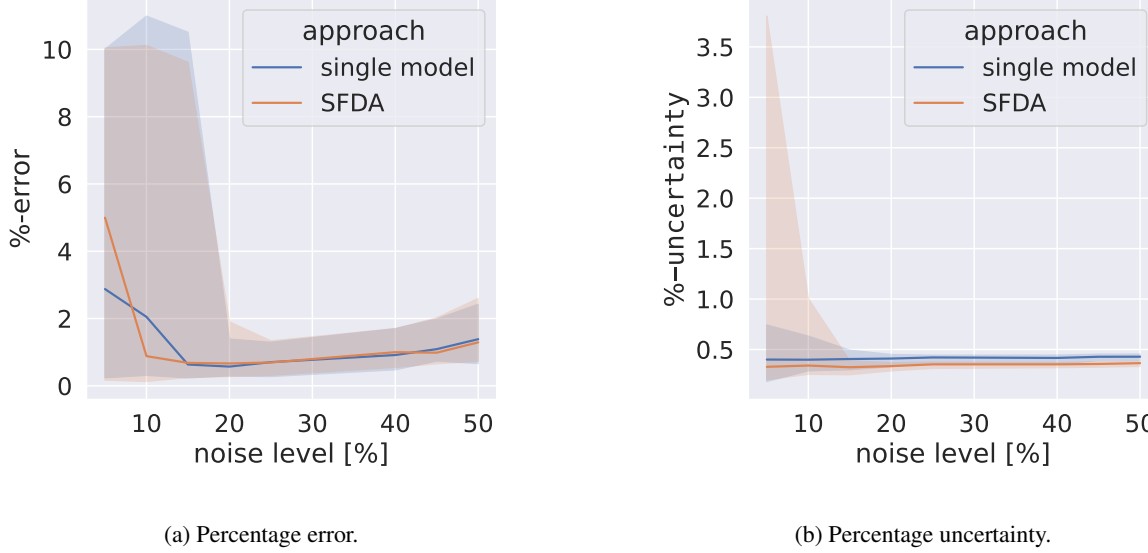

(a) Percentage error.

(b) Percentage uncertainty.

**Figure 6.** The percentage error between the true values of $(\sigma, \rho, \beta)$ and those from SFDA, as well as average percentage uncertainty on the SFDA parameters. A single model assimilation is included for comparison. An ensemble of 100 assimilations is carried out over 100 different data sets. The median (line) and $68\%$ percentile intervals (shaded areas) are plotted. The noise level varies between 5% and 50% in steps of 5%.

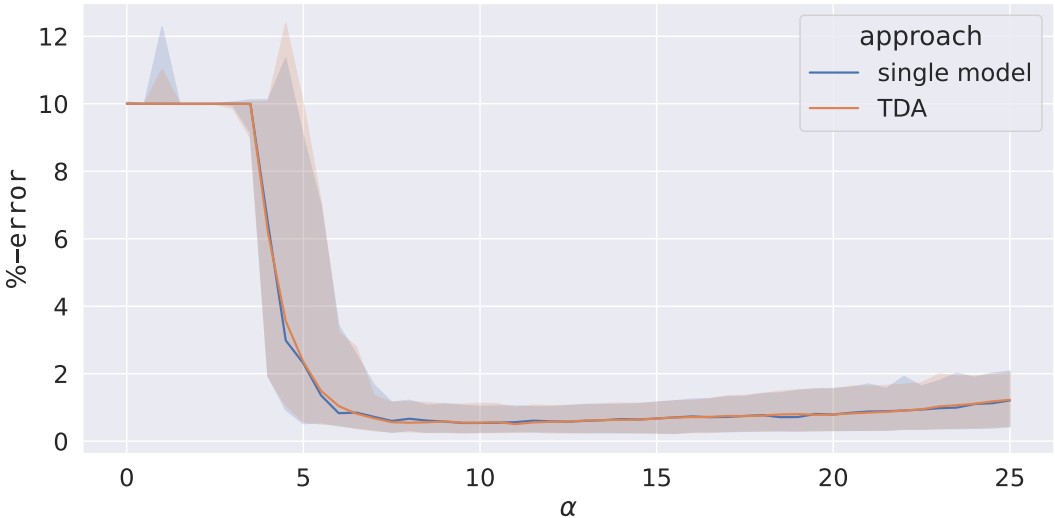

**Figure 7.** The percentage error between the true values of $(\sigma, \rho, \beta)$ and those from TDA. A single model assimilation is included for comparison. An ensemble of 100 assimilations is carried out over 100 different pseudo-data sets. The median (lines) and $68\%$ percentile intervals (shaded areas) are plotted. The noise level is $25\%$.

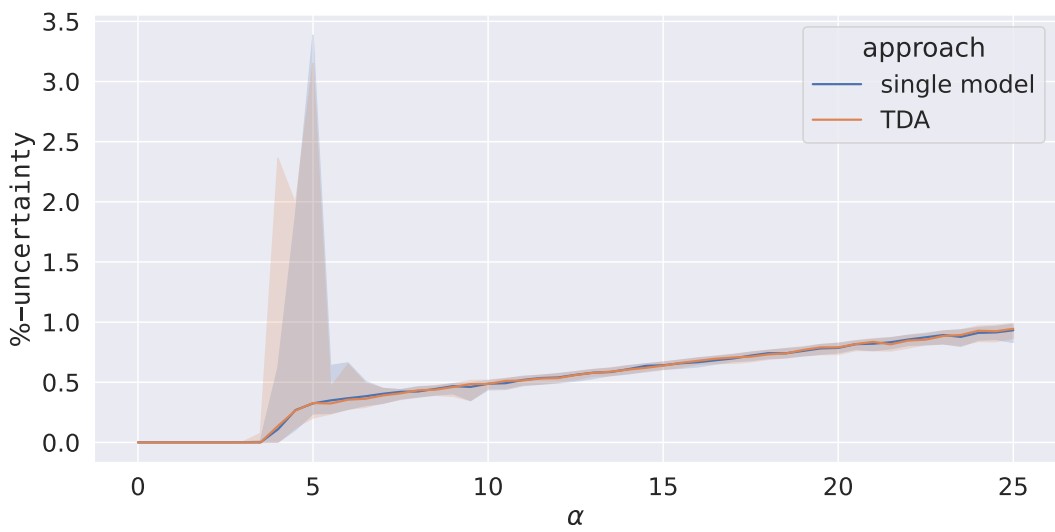

**Figure 8.** The percentage uncertainty, from the minimisation algorithm, averaged over all three parameters $(\sigma, \rho, \beta)$ after TDA. A single model assimilation is included for comparison. An ensemble of 100 assimilations is carried out over 100 different data sets. The median (lines) and 68% percentile intervals (shaded areas) are plotted.

Fig. 9 shows the results of varying the noise levels on the fitted parameter values. For all noise levels studied, the fit can be considered to be accurate as the mean percentage error on the parameters remains below 1% even with noise levels of 50%. The increased spread of the error at low noise is thought to be due to the fixed value of $\alpha$ used for all noise levels impacting the synchronisation of the system. The TDA setup is found to have extremely consistent uncertainty compared to the single model system.

The consistent results of TDA in Figs. 7, 8, and 9 relative to the standard single model setup show that transferring information via synchronisation does not compromise precision. Figs. 7, and 8 also concur with those of SFDA in suggesting that the optimal value of $\alpha$ is the smallest value after the onset of effective synchronisation. An increase in $\alpha$ beyond this point can lead to a significant reduction in the precision of the parameters. In cases where only a simpler, but similar, model with an adjoint is available results are likely to degrade. In the following section, we will study the potential impact of model inconsistencies

on the the precision of the parameter estimation.

### 4.3    Mismodelling in TDA

In this section, the tandem data assimilation (setup 2) is be used with different forward and adjoint models that share common physics to examine the impact of introducing model discrepancies. We construct a test case where the equations of model 2, which has an adjoint, Eq. 8 are modified to give an oscillatory difference to both the true model and model 1. This can be done

in a number of ways. We choose to introduce a multiplicative sine function to the equations in such a way it is also included in

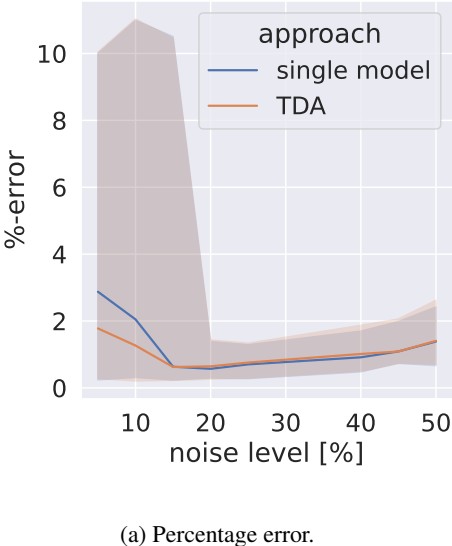

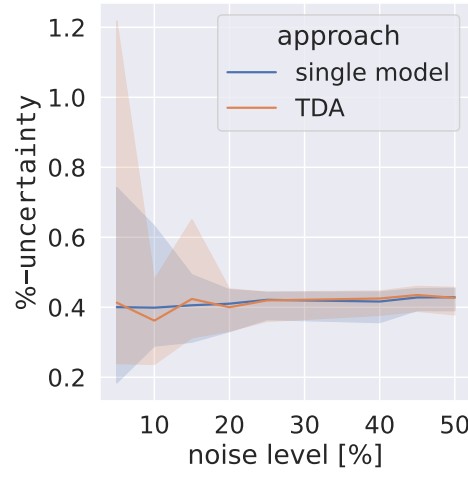

(a) Percentage error.

(b) Percentage uncertainty.

**Figure 9.** The percentage error between the true values of $(\sigma, \rho, \beta)$ and those from TDA. A single model assimilation is included for comparison. An ensemble of 100 assimilations is carried out over 100 different data sets. The median (lines) and $68\%$ percentile intervals (shaded areas) are plotted. The noise level varies between 5% and 50% in steps of 5%.

the adjoint matrix and thus modifies the gradient values returned to the fitting algorithm. Model 2 with an adjoint is now

$$\frac{dx_a}{dt} = \sigma(y_a - x_a) + \alpha(x_f - x_a), \tag{19a}$$

$$\frac{dy_a}{dt} = \rho x_a - y_a - x_a z_a + \alpha(y_f - y_a), \tag{19b}$$

$$\frac{dz_a}{dt} = x_a y_a - \beta z_a \cdot (1 - \epsilon \sin(2\pi t)) \tag{19c}$$

where $\epsilon$ is term which determines the strength of the oscillation term. The effect of this term on the attractor is shown in Fig. 10, without synchronisation. When compared to Fig. 2a, it can be seen that this term is successful in distorting the shape and probability density of the attractor.

The consequences of varying this term on the accuracy of parameter optimisation after assimilation are shown in Fig. 11. With increasing $\epsilon$ the percentage error and uncertainty between fitted and true systems remains stable. In spite of the large 360 impact this term has on the attractor shape, the figure demonstrates a resilience of TDA to modelling differences between the forward-only model 1 and model 2 with an adjoint.

## 5 Conclusions

In this paper we have demonstrated the ability to constrain a Lorenz '63 model using a second model with similar physics and an adjoint by 4D-Var data assimilation. Such an approach removes the need to generate an adjoint for a forward model,

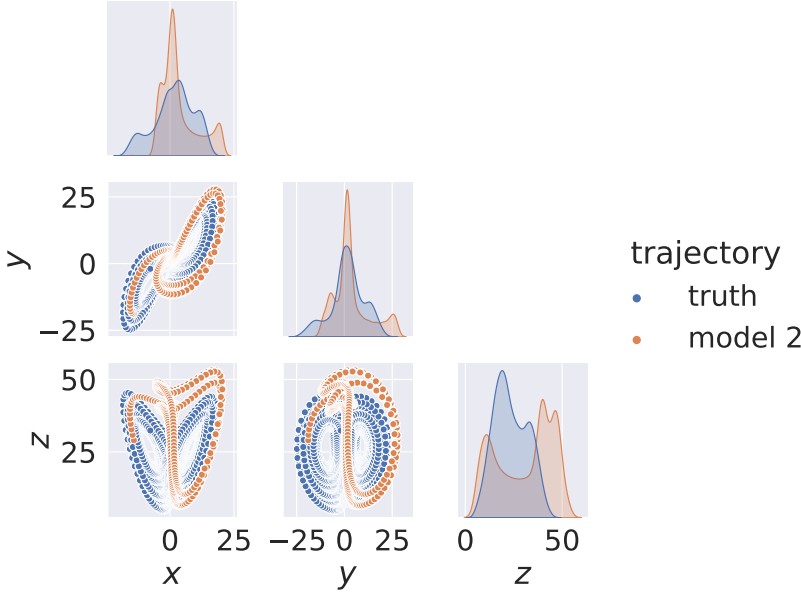

**Figure 10.** The Lorenz true model and model 2 attractors in the case of $\epsilon = 1.0$. The bottom left and top right quadrants shows the attractors from all possible co-ordinate orientations. The diagonal plots show kernel density estimations (KDEs). No noise is added.

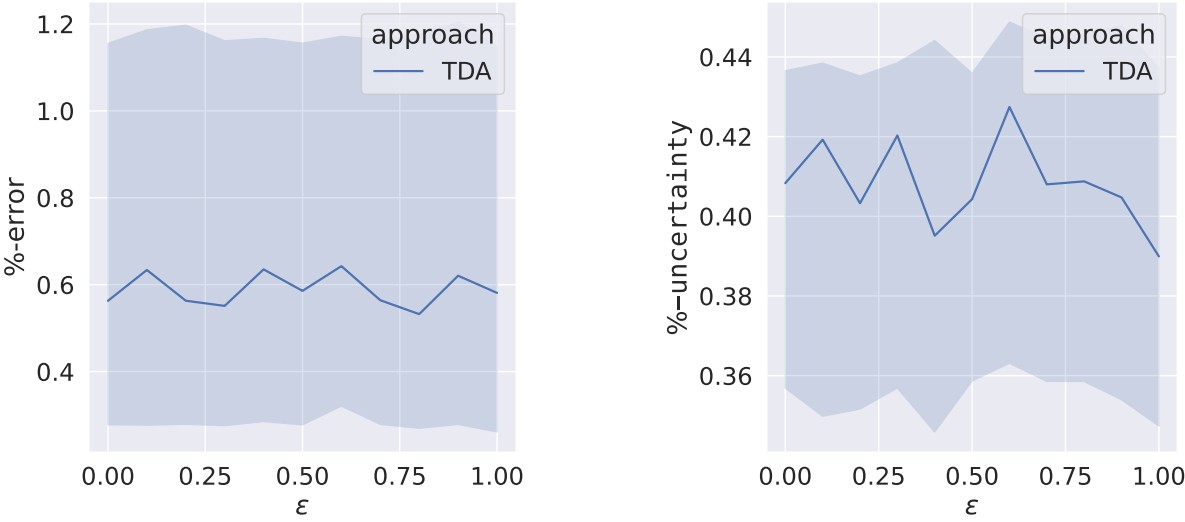

**Figure 11.** The percentage error (left) and uncertainty (right) between the true values of $(\sigma, \rho, \beta)$ and those from TDA. An ensemble of 100 assimilations is carried out over 100 different data sets. The median (line) and $68\%$ percentile intervals (shaded areas) are plotted. The noise level $25\%$ and, $\alpha = 7.5$.

if such an adjoint already exists for a separate, yet dynamically similar system. An important application of this technique in Earth system modelling would be a situation where a low-resolution ESM with an adjoint shares a parametrisation with a high-resolution ESM for which no adjoint exists. Moreover, using a lower-resolution version of the same model could computationally make data assimilation much faster. We have shown that in both cases the low-resolution ESM with an adjoint could, through synchronisation, follow the trajectory of the more complex and high-resolution model while at the same time

providing all necessary variables to run its tangent linear adjoint model. This can then be utilised to estimate parameters in the complex high-resolution ESM. We have also shown that running a forward model twice before beginning data assimilation can act to smooth the data and reduce the parametric uncertainty. Our focus is in optimising the parameters of a full ESM, which will be tested as a next step. It would also be possible to optimise the initial condition of the state variables, which is more applicable to weather forecasting techniques. Future work will examine the resilience of such setups to spacially and

temporally sparse data.

*Data availability.* The pseudo-data samples used in this study are available on request from the authors.

*Author contributions.* PDK carried out the research, wrote the initial manuscript draft, and edited it. AB assisted with the research and edited the manuscript. AK and DS designed the research concept, supervised the work, and participated in the writing of the manuscript.

*Competing interests.* The authors declare no potential competing interest.

*Acknowledgements.* The authors would like to thank the two anonymous referees for providing very detailed and constructive reviews which significantly contributed to the final form of this manuscript. The figures in this research were plotted by the `seaborn` plotting package (Waskom, 2021) utilising our output which was managed using `pandas` (pandas development team, 2020). Automatic differentiation of our model was carried out using `JAX` (Bradbury et al., 2018). The minimisation of our cost function and uncertainty evaluation was done by `iminuit` (Dembinski and et al., 2020). This research was supported, in part, through the Koselleck grant *Earth^{RA}* funded by the Deutsche

Forschungsgemeinschaft (grant number: 66492934.) This work used resources of the Deutsches Klimarechenzentrum (DKRZ) granted by its Scientific Steering Committee (WLA). Contribution to the Centrum für Erdsystemforschung und Nachhaltigkeit (CEN) of Universität Hamburg.

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
