# Peer review of "Long-window tandem variational data assimilation methods for chaotic climate models tested with the Lorenz 63 system"

_EGUsphere, 2024_

## Referee Comment (RC2)

Review of *Long-window hybrid variational data assimilation methods for chaotic climate models tested with the Lorenz 63 system* by Kennedy, Banerjee, Köhl, and Stammer, submitted to NPG.

Kennedy and co-authors present a new and interesting method of parameter estimation for Earth system models, in particular to estimate parameters for a high-resolution model for which there is no adjoint, using a lower-resolution model of the same dynamics for which an adjoint is present. Their technique uses "synchronization" of these two dynamical models using pseudo-observations to estimate the parameters. The authors derive and test this technique on the Lorenz '63 model. The goal of synchronization is to improve the length of the time integration of the high-resolution model through improved parameter estimation, with applications to climate modeling, which run forecasts on longer time scales than numerical weather prediction.

The authors present a technique that may be of use for climate modeling applications, and I can easily see the extension of this technique to more complex models. The foundations of this paper are good, however, there are several parts of the paper that require further explanation, clarification, clear definitions, and contextualization in the current data assimilation literature. Addressing these questions and comments will significantly improve the manuscript and its contribution to the community. Therefore, I recommend major revisions for this manuscript before being considered for publication by NPG.

**Major Comments**

I begin with a series of major comments, which if addressed, can significantly improve the clarity and purposes of this paper.

1. My first major comment is necessary to address, because it will clarify the contributions of this paper. If I understand correctly, the goal of this paper is to estimate the parameters of a dynamical model that will be used for forecasting the states of this dynamical model for long time periods (i.e., on climate timescales). This dynamical model does not have an adjoint, therefore a variational data assimilation approach for estimating these parameters given observations cannot be done. However, a simpler, related dynamical model does have an adjoint, therefore optimization with this adjoint can be used to estimate these parameters, which is does through a process the author's call "synchronization." If this is correct, then this needs to be clarified in the introduction and Section 2. Below are a series of more specific details regarding this comment:

   - In the second paragraph of the introduction and first paragraph of Section 2.2, the authors refer to a "cost function," however, having an explicit formula for this cost function, particularly in Section 2, will help to clarify the author's intention. This will emphasize the need for an adjoint (as well as define the adjoint prior to its definition in Eq.(2)), clearly define the arguments of the cost function for which you intend to minimize, and contextualize this work within the existing variational data assimilation literature. In addition, it would be helpful to clarify whether you are also minimizing such cost function for the state estimate as well, therefore defining a joint state-parameter estimation problem. For example, Chapters 4 and 5 of Evensen et al. (2022), formulate weak constraint and strong constraint 4D-Var data assimilation for the joint state parameter vector **z**. It would be very helpful to compare what you are doing with standard formulations, such as those presented in this book. With respect to the cost functions defined in 2.4.1-2.4.4, these cost functions look different than the standard 4D-Var cost functions in the data assimilation literature (e.g. like those presented in Desroziers et al., 2014; Evensen

et al., 2022). The authors should explain the difference between these cost functions and the cost functions used in 4D-Var, which again will help to clarify the intentions of this work and contextualize it within existing data assimilation literature.

- This next comment is regarding the specific details of the experiments: In the first paragraph of Sec. 2.2, the authors use the phrase, "control parameters," however it is unclear if these are are these the model parameters $\sigma, \rho, \beta$ or possible the state variables $x, y, z$. Definition of a cost function in this section would address this question. Second, is this set up correct: the assimilation window is 100 model time units, and over this window only the parameters of the Lorenz '63 are estimated (the state variables $x, y, z$ are not), and this generates estimates of new parameters? What is the frequency of the pseudo-observation time series that is assimilated in this window? After the new parameters are estimated, do the authors perform a forecast of the state with these new parameters to compare with the true model to compute the RMSE? The content of Sections 2 and 3 can be expanded to address these questions, which will help the readers better understand the experiments. This will also help to clarify results presented in figures in Sec. 4.

- The authors introduce the idea of synchronization: in the abstract, there are facts about synchronization that are described in the abstract (such as reducing positive Lyapunov exponents to negative values) that should also be discussed in Section 2.3, and possibly in the introduction as further motivation for this technique. I suggest adding a more detailed description of synchronization in the beginning of Sec 2.3, particularly after the sentence "The problem can be mitigated by synchronization..." Are there any simple examples that can illustrate the synchronization technique one could describe here, before showing how it applies to the Lorenz '63 system?

2. The second major comment I will make is on the literature review and discussion of data assimilation, which begins in the first two paragraphs of the introduction and is discussed in various places throughout the rest of the manuscript. In order to correctly contextualize and understand the contributions of this work, the authors can expand their literature review on data assimilation. In the second paragraph of the introduction, the authors state that there are two common approaches to data assimilation, "sequential data assimilation" and "variational approach." This is correct, but can improved. Sequential data assimilation should be explained and contrasted with variational data assimilation: if by sequential data assimilation you mean Kalman filters and their variations (e.g, extended Kalman Filter, ensemble Kalman filters, square-root filters), please specify these and cite the appropriate references (for instance, but not limited to Kalman, 1960; Evensen, 1994; Tippett et al., 2003; Houtekamer and Mitchell, 2001) and note that these methods, by design, are adjoint free methods since they do not require minimizing a cost function. With respect to the variational methods, there are other variational schemes in addition to 4D-Var, such as 3D-Var, weak-constraint and strong constraint 4D-Var, and the ensemble-variational filters such as 4DEnVar (Desroziers et al., 2014, and references therein). A brief review of these methods and references should be included.

3. The use of the word "hybrid data assimilation" particularly in the title and the method introduced in Sec. 2.4.2, either needs to be changed or properly contextualized, as it may be a bit misleading. The term "hybrid data assimilation" exists in the data assimilation literature, first defined in Hamill and Snyder (2000), where the term "hybrid" arose from combining technique from 3D-Var and the EnKF to incorporate flow-dependence in the background

estimation error covariance in the 3D-Var update step. To my understanding, the authors here are not performing hybrid data assimilation, but rather introduce a hybrid of their two models for the parameter estimation. To avoid confusion, I request that the authors change this terminology or clearly distinguish what they mean by hybrid from the terminology that already exists in the data assimilation literature.

**Minor Comments**

- In the second sentence of the introduction that beings with "ESMs can be used ..." Even though ESM is defined in the abstract, it also needs to be redefined in the text and should not start a sentence. This can be rewritten as "Earth system models (ESMs) can be used ..."

- The word "precision" is used throughout (e.g., last sentence of the abstract, fourth paragraph of the introduction, twice on page 4, once on page 11, twice on page 12). I believe the authors mean to use "accuracy" instead, since the authors are looking at errors compared to a truth run. Precision is defined as repeated experiments yielding the same result, though that result may not be accurate.

- In the first sentence of Sec. 2.1, please cite the original paper Lorenz (1963) in addition to the Yang et al. (2006) paper.

- Regarding the pseudo-observations in Sec. 2.2, (1) how often are these observations saved in the time series, and (2) when defining the additive Gaussian noise, can you please specify the mean and variance for the white noise (this may be defined in the last sentence of this paragraph, however this is a bit unclear)?

- In Eq. (2): please define the vector $\boldsymbol{x}$; I assume this means $\boldsymbol{x} = (x, y, z)$, however this should be defined explicitly either before, along with, or immediately after Eq. (2). The same goes for the vectors $\boldsymbol{x}_0$ and $\boldsymbol{x}_a$ in Eq. (7) and $\boldsymbol{x}_f$ in Eq. (9).

- The beginning of Sec. 2.4: what do you mean by "This" in the first sentence? Can you briefly summarize what you mean by this?

- Regarding Fig. 1, the figure looks nice, however making the linewidths thicker will make it a bit more readable.

- Section 2.4.2, the sentence "...may differ in resolution or numerical formulation by are governed by the same equations" By "same equations" do you mean same continuum dynamics?

- Regarding Eqs. (8) and (10), it is unclear where these adjoints appear. To improve this, I suggest the authors reorganize section 2.4 by combining sections 2.4.1 with 2.4.4 and 2.4.2 with 2.4.5 so that these equations are adjacent in the text and better explain your methodology to the readers.

- In the second paragraph of the Results, why do the authors plot the 68% percentile? A brief sentence justifying this choice would be useful.

- In the first second paragraph of Sec. 3.1, the authors say "results of two fits carried out with noise of 25%" can you clarify what it is meant by "noise of 25%" namely what is this noises added to, and what is the 25% being applied to calculate the noise.

- This is a question that may be of interest: based on your experiments, is there an optimal choice of $\alpha$? For example, Fig. 7 is slightly concave up, indicating a value of $\alpha$ that can minimize the error. This may be worth exploring.

- The first sentence of Sec. 3.3 begins with the acronym "HDA," I suggest that the authors change this sentence so that it does not start with an acronym.

**References**

Desroziers, G., Camino, J., and Berre, L. (2014). 4DEnvVar: link with 4D state formulation of variational assimilation and different possible implementations. *Q. J. Roy. Meteor. Soc.*, 140:2097–2110.

Evensen, G. (1994). Sequential data assimilation with a nonlinear quasi-geostrophic model using Monte Carlo methods to forecast error statistics. *J. Geophys. Res.*, 99:10143–10162.

Evensen, G., Vossepoel, F., and van Leeuwen, P. (2022). *Data Assimilation Fundamentals*. Springer.

Hamill, T. M. and Snyder, C. (2000). A Hybrid Ensemble Kalman Filter–3D Variational Analysis Scheme. *Mon. Weather Rev.*, pages 2905–2919.

Houtekamer, P. L. and Mitchell, H. L. (2001). A sequential ensemble Kalman filter for atmospheric data assimilation. *Mon. Weather Rev.*, 129:123–137.

Kalman, R. (1960). A new approach to linear filtering and prediction problems. *Journal of Basic Engineering*, 82:35–45.

Lorenz, E. (1963). Deterministic nonperiodic flow. *J. Atmos. Sci.*, 20:130–141.

Tippett, M., Anderson, J., Bishop, C., Hamill, T. M., and Whitaker, J. (2003). Ensemble Square Root Filters. *Mon. Weather Rev.*, pages 1485–1490.

---

## Author Comment (AC3)

[revised manuscript text omitted]

$$M^T \equiv \frac{\partial \dot{\boldsymbol{x}}_i}{\partial \boldsymbol{x}_j} = \begin{pmatrix} -\sigma & \sigma & 0 \\ \rho - z & -1 & -x \\ y & x & -\beta \end{pmatrix}.$$

**3.3**

~~A fundamental limitation of the adjoint method arises when integrating over periods that are longer than the predictability time scale of a system, leading to exponentially growing gradients and a cost function with an increasing number of local minima (Köhl and Willebrand, 2002; Lea et al., 2000). The problem can be mitigated by synchronisation which removes the non-linear or chaotic dynamics which effect the cost function (Abarbanel et al., 2010; Sugiura et al., 2014). To incorporate a synchronisation technique, weby adding so-called nudging~~ (
[revised manuscript text omitted]
 subsequently present. The adjoint model is numerically evaluated by automatic differentiation (AD) of the forward model 2 with respect to either model 1 or 2 depending on our~~

chosen setup. This is done in the python package `JAX`     `JAX`.     `iminuit`  `Minuit2`  `MIGRAD`

**3.3.1 Setup 1 - state filtered data assimilation (SFDA)**

Assuming that both  model 1 and 2 can be thought of as representing two identical climate models, the cost function can be placed on the model 2. This allows model 1 to filter out some of the background noise on the observations before they are given to the cost function attached to model 2.   Such a filtering setup would theoretically reduce parametric uncertainty below that of traditional single model data assimilation because model 1 should act to reduce the amount of noise synchronised into model 2. We will subsequently refer to this setup as state filtered data assimilation (SFDA.)

In SFDA the cost function acts to constrain model 2. The cost function is

[revised manuscript text omitted]

**3.3.3  Cost function gradient**

The gradient of the cost function, with respect to the state variables at $t = 0$, is given by

$$
\nabla_{x_a} J = \frac{1}{N} \lambda(0),
$$

where $\lambda$ is the adjoint vector. The adjoint vector is calculated by integration of the differential adjoint equation in the reverse time direction. This will differ between our two setups.

**3.3.3  SFDA (setup 1)**

This adjoint equation for SFDA is given by:

$$
\dot{\lambda}_{\text{SFDA}}(t) \equiv \frac{1}{\sigma_{x_o}^2} (x_o(t) - x_a(t))
$$

$$
- M_{\text{SFDA}}^*(t) \lambda_{\text{SFDA}}(t) \text{ for } t = N, ..., 0
$$

with $\lambda_{\text{SFDA}}(N) \equiv 0.$

These equations were derived using the method detailed in Talagrand (2010). The gradient can then be calculated with respect to the parameters $(\sigma, \rho, \beta)$ notated by the subscript $\theta$. This yields

$$
\nabla_{\theta} J_{\text{SFDA}} = \frac{1}{N} \sum_{t=N}^{0} \lambda_{\text{SFDA}}(t) \begin{pmatrix} y_a(t) - x_a(t) \\ x_a(t) \\ -z_a(t) \\ y_f(t) - x_f(t) \\ x_f(t) \\ -z_f(t) \end{pmatrix}
$$

which is a component-wise multiplication at each time step.

**3.3.4**

 The adjoint equation for  TDA is given by:

$$\dot{\boldsymbol{\lambda}}_{\underline{\mathrm{HDA}}\,\mathrm{TDA}}(t) = \frac{1}{\sigma^2_{\boldsymbol{x}_o}}\left(\boldsymbol{x}_o(t) - \boldsymbol{x}_f(t)\right) \tag{15a}$$

$$-\boldsymbol{M}_{\mathrm{HDA}} - \boldsymbol{M}_{\mathrm{TDA}}{}^*(t)\boldsymbol{\lambda}_{\underline{\mathrm{HDA}}\,\mathrm{TDA}}(t) \text{ for } t = N,...,0$$

with $\boldsymbol{\lambda}_{\underline{\mathrm{HDA}}\,\mathrm{TDA}}(N) = \boldsymbol{0}.$ \hfill (15b)

The gradient with respect to the parameters $\boldsymbol{\theta} = (\sigma, \rho, \beta)$ is calculated to be:

$$\boldsymbol{\nabla}_{\boldsymbol{\theta}}J_{\underline{\mathrm{HDA}}\,\mathrm{TDA}} = \frac{1}{N}\sum\int_{t=N}^{0}dt\,\boldsymbol{\lambda}_{\underline{\mathrm{HDA}}\,\mathrm{
[revised manuscript text omitted]

---

## Author Response (AR1)

Response to anonymous referees

April 18, 2025

We would like to thank both referees for their extremely constructive comments. During the revision process, we took all of them into consideration, following a colour coding system explained below. The manuscript has improved considerably thanks to these comments and we hope that it is now ready for publication.

**Colour code:**

- Anonymous referee comment.

- Comment agreed and resolved.

- Response to comment and reasoning for not making the suggested change.

**1 Referee 1 major comments:**

1. In general, I feel the authors do not clearly state the status in the field of parameter estimation in climate models and the motivation of the study. For example, authors should clearly state the rationale behind the proposed framework and the problems being solved. Why do you need a long DA window for parameter estimation? Why do you think the new framework have any benefits? In the current formulation, the parameter estimation also requires a sensitivity matrix to the parameters. Would this be difficult to be obtained in complex climate models?

The abstract has been modified and the introduction has been expanded taking into account this comment and those of referee 2. For an answer to the questions raised here please see point 4 of the detailed comments.

2. The methodology section needs significant restructure. The authors should first briefly introduce the synchronisation method as a generic method instead of its L63 formulation. Then, the cost function of variational method should be introduced along with the gradient of the cost function. Finally, the authors should introduce the Lorenz 63 model, the exact formulation of the synchronisation model with the Lorenz 63 model, and terms in the cost function and its gradient Lorenz 63 model. The paper might also be benefited from an experiment setup (sub)section, which provide details of the choice of nudging strategy, the chosen value of observation noises/twin experiments setup, the metrics used etc. One of the given benefits of synchronisation approach is the possibility of using long DA window. However, the DA window of 100TUs is given at the very end of Sect. 3.2.

The methodology section has been reorganised following the above suggestions. A separate experimental setup section has also been added to introduce the Lorenz '63 model specific details.

3. The authors need to check Eq. (7 - 15). The adjoint equations presented in this study are normally obtained when the cost functions are temporal integral because it involves integration by parts. Yet, the cost functions are given as discrete time summations. In the SFDA section, the cost function is given as the misfit between observations and $x_a$, presumably $x_a$ has 3 elements. However, given Eq (8), the gradient of the cost function in Eq. (12) does not hold as Eq. (12a) is a differences between a 3 element vector and 6 element vector as $M*_{SFDA}$ is a 6 x 6 matrix. Do authors define $x_a = (x_f x_a)$, or is $M*_{SFDA}$ given incorrectly? Moreover, it is unclear to me why do authors decide to provide the gradient of cost function (Eq. 13, Eq. 15) with respect to the parameter theta as an element-wise multiplication while it is supposed to be a matrix-vector product of dM/dtheta and lambda(t).

All equations have been double-checked and modified where required, and integrals are now used in place of sums to be consistent with the text book terminology on the adjoint method.

4. More interpretation is needed for results section. In Figure 4 and 7, with small alpha, the solid line does not look like the median of the ensemble, and I can only guess that all results lead to increased errors. Is this the case? Also, why does the parameter estimation perform better with increased alpha? Authors also need to provide better comparison and interpretation for the differences between the single, SFDA and HDA

approach. What causes the need for different alpha? The lack of interpretation leads to very similar Sect. 3.1 and 3.2.

We followed this advise by adding more interpretation of the presented results and by expanding our discussion about optimal values of $\alpha$.

5. It will be good to look at the impact of different length of DA window on the parameter estimation, or Lyapunov exponents of the system. It may also be useful to check the performance when components of the syncrhonised system have different parameter values.

The Lyapunov exponent of the Lorenz '63 model is directly dependent upon its parameters. By varying the parameter through the optimisation, we investigate a wide range of scenarios. For perfect models with additive noise, the precision of the recovered parameters will improve with increasing window length, as a result of the law of large numbers. Comparing estimates obtained over long windows with averages from estimates over short windows, for which no synchronisation is necessary, would be a very valuable addition to the paper. However, short window assimilations could be done in various ways. Windows could be treated independently and results could be averaged, or (probably more naturally) optimisations could be performed sequentially using the results from the previous window as background information. This would require more detailed work, which would expand the paper quite a bit. However, we added respective discussions (see answer to detailed comment 2 below.) We covered the case of an imperfect model (similar to systems having different parameter values) by perturbing the equations with extra terms. Since all parameters are optimised, they can only be different if we reduce the number of optimised parameters, and make the remaining parameters different. As the system has only three parameters, we did not want to make optimisation even simpler.

**2    Referee 1 detailed comments:**

1. "a sequential data assimilation scheme (Bertino et al., 2003) and the variational approach (Le Dimet and Talagrand, 1986)." $\rightarrow$ "...sequential and variational data assimilation schemes..."

In paragraph 2 of the introduction: 'There are two common assimilation approaches typically used to incorporate observations into a model: sequential and variational data assimilation schemes (Wunsch, 1996).'

2. "defined as the quadratic misfit between the observational and model data within an assimilation time window" $\rightarrow$ Usually, 4DVar cost function has a background term making it equivalent to a maximum likelihood problem in the view of Bayesian theorem under Gaussian assumption. It would be useful to distinguish the cost function here compared to more common cost function formulation

It is true that a background term is usually employed, since prior information is usually available and since it is also required to guarantee well-posedness of the problem. However, since we are confident that the observability of the parameters are not a problem we consider the limit infinitely small weights on the background term such that the background term does not affect the estimation of the parameters.

The formulation of the problem follows the strong constrained formulation described in Chapter 4 of Evensen et al. (2022) where the primary goal is to estimate unknown model parameters, and the secondary goal is to improve its state through said parameters. Therefore we are using joint state-parameter estimation problem without a joint vector $z$ as in Evensen et al. (2022). Different from their formulation no background term is employed. Even though prior information is available and it would guarantee a well-posed problem, we consider the limit of infinitely small weights on the background term, such that it does not affect the estimation of the parameters. We don't think this is a problem as we are confident in the observability of the parameters. We added this detail.

3. "Due to the nonlinearities within ESMs..." $\rightarrow$ "The use of adjoint models face several challenges. Due to..."

'In the context of a full non-linear ESM, the use of adjoint models face several challenges.'

4. "the problem can be mitigated by synchronisation which removes the non-linear or chaotic dynamics from the adjoint model leading to a smooth cost function" – I feel it might be better to phrase it as "...synchronisation which constructs a system with reduced sensitivity to initial conditions leading to ..."; Moreover, authors should discuss the benefits of long DA window especially for parameter estimations.

In the first paragraph of section 2.1 we now state: 'The non-linear or chaotic dynamics, which detrimentally effect the maximum likelihood estimate, can be removed by synchronisation (Abarbanel et al., 2010; Sugiura et al., 2014) which transforms the chaotic model into one with linear dynamics without positive Lyapunov exponents leading to maximum likelihood functions with one unique maxima.'

We also added the following text to the second paragraph of section 2.1 to address the benefits of long

window DA: 'According to the law of large numbers both with perfect models and in the presence of noise, the precision of the recovered parameters will improve with increasing window length since more data is integrated into the estimation. Similar benefits could be achieved by averaging estimates obtained over short windows, for which no synchronisation is necessary. However, underlying restrictions differ. For synchronisation, noise affects the state over the entire window, whereas for short windows noise effects are transported. Short window assimilation can be of benefit in perfect model settings from the error growth as suggested by the quasi-static variational assimilation (QSVA) framework (Pires et al., 1996) due to fact that sensitivities increase exponentially with time in chaotic models. The analogue of this QSVA effect in the Dynamical State and Parameter Estimation (DSPE) method (Abarbanel et al., 2009) is the attempt to reduce the synchronisation parameter as the optimisation progresses and parameters move closer to their true values. Since errors and sensitivities grow exponentially, feasible window lengths in QSVA have a maximum value due to limited numerical precision. Similarly, synchronisation parameters cannot approach zero for assimilation windows much larger than the predictability limit, because synchronisation will eventually fail if positive Lyapunov exponents exist (Quinn et al., 2009). We note that the reasoning for the need of long assimilation windows is somewhat different in the context of full ESM, for which it is essential to resolve long time scale physical mechanisms impacted by the specific choice of parameters, such as air-sea interactions of advection time scales in the ocean.'

5. "This method allows extension of" – "This method allows for the extension of"

In the fifth paragraph of the introduction we now say: 'This method allows for the extension of the assimilation window beyond the predictability time-scale, provided that sufficient observations are available.'

6. "To mitigate both problems, we propose a novel framework where we use two climate models both coupled through synchronisation, one with a high resolution and the other with coarse resolution for which an adjoint exists. " – Here, is it the common that adjoint models of coarse resolution is available while the adjoint models of high-resolution models are not available? Authors should provide references and discussions on the existence of the issue. Further, authors should also discuss how this novel framework could mitigate the problem of smoothness and dimensionality.

We added the following information to paragraphs six and seven of the introduction: 'The creation of an adjoint model code from the forward code usually requires considerable effort. Automatic differentiation tools, such as Giering and Kaminski (1998); Hascoet and Pascual (2013) were developed to aid in this step. But substantial changes to the forward model code are required unless it was already developed with the adjoint modelling in mind. Stammer et al. (2018) created the first adjoint of an intermediate complexity fully coupled earth system model that is automatically created from the forward model by automatic differentiation using the TAF compiler, called the Centrum für Erdsystemforschung und Nachhaltigkeit (CEN) Earth System Assimilation Model (CESAM). The adjoint of this intermediate-complexity model is intended to be utilised for tuning more complex CMIP-type models through parameter estimation since the basic underlying physics is very similar. Otherwise this is a manual process with considerable ambiguity in the choice of parameters (Mauritsen et al., 2012).

Therefore, we propose a novel framework in which we use two climate models both coupled through synchronisation, one with a high complexity and the other of intermediate complexity for which an adjoint exists to address the second problem. The technique also has a much wider range of additional applications, since resolutions using the adjoint method lag behind those applications featuring simpler assimilation methods as variational methods are typically a factor of 100 more costly than running the associated forward model. For example, the global GECCO3 ocean reanalysis based on the adjoint method (Köhl, 2020) features only a nominal resolution of 0.4°, while for instance the GOFS 3.1 (Laboratory, 2016) based on 3D-Var (Cummings and Smedstad, 2013) features 1/12°resolution. Employing coarser versions of the adjoint while still running the forward model with full resolution could significantly reduce the cost of the assimilation effort. Therefore, the objective of this paper is to investigate the accuracy and precision of such a synchronised data assimilation approach. We perform this test using Lorenz '63 model.'

7. "The objective of this paper is to quantify the precision and the benefit of such a synchronised data assimilation approach." – I believe this is what you have done instead of the objective of the study. A better objective would be to investigate the performance of the novel approach you proposed.

Combination of referee comments 1 and 2 in paragraph seven of the introduction: 'Therefore, the objective of this paper is to investigate the accuracy and precision of such a synchronised data assimilation approach.'

8. "We perform this test conceptually using a Lorenz 63 model system." – The test is not performed "conceptually".

In paragraph seven of the introduction:'We perform this test using a Lorenz '63 model.'

9. "The advantage is that it can be used to quantitatively evaluate the parameter dependence of the system prior to application in a full model." – This sentence needs rephrasing. I guess the authors want to say "...quantitatively evaluate data assimilation schemes..." because the parameter dependence of a system will change for different dynamical systems. In fact, parameters of Lorenz 63 are non-dimensionalised numbers of a convection system. These parameters may not appear explicitly in a full climate model.

*Paragraph eight of the introduction now states: 'The advantage is that it can be used to rapidly evaluate parameter estimation techniques in data assimilation schemes prior to their application in a full ESM with low computational resource requirements.'*

10. "It can also be used in a wide range of other applications (Du and Shiue, 2021; Cameron and Yang, 2019; Pelino and Maimone, 2007). " – you might want to describe examples of these applications.

*Paragraph eight of the introduction now states: 'It can also be used in a wide range of other applications including, but not limited to, data assimilation, stochastic modelling terms, and predictions (Du and Shiue, 2021; Cameron and Yang, 2019; Pelino and Maimone, 2007).'*

11. Eq. (1) describes the classic L63 model, which I believe should have a reference to the original paper by Lorenz.

*Section 3.1 now opens with: 'In this study, we use the Lorenz '63 system for all our experiments (Lorenz, 1963).'*

12. "Sub-section" can be just "Section"

*'Section'*

13. "The random noise value magnitudes are bounded by a given percentage relative to the systems' standard distribution." — Please provide more details of your random noise choices. This is for the sake of reproducibility and credibility of the research.

*Section 3.1 now includes: 'The random noise magnitudes are bounded to 25% of the Lorenz '63 system's standard deviation.'*

14. Eq. (2) is technically not the adjoint model/TLM. The TLM is defined as $d\delta x/dt = M\delta x$. Also, matrices are conventionally given as bold capital letters and the matrix transpose operator should not be italic. The vector x as well as the dot operator is not defined here. In fact, I doubt the necessity of this equation as this study does not use this equation at all.

*As recommended these equations are now removed.*

15. In Eq. (3), $x_a, y_a, z_a$ are not defined. Considering that authors discuss the nudging of the $z$ variable, would it be good to have $\alpha(z_o - z_a)$ term in Eq. (3) first? Moreover, this is an equation of synchronisation strategy specifically for Lorenz 63 model. Could the authors provide a general description of synchronisation before case-specific description? Also, is this the single model approach mentioned in Sect. 3? If this is the case, authors should clearly state it.

*All points raised here have been resolved by the re-ordering of the methodology and experimental setup sections.*

16. Again, I doubt the necessity of having Eq. (4).

*Removed.*

17. "...hybrid data assimilation (HDA.)..." $\rightarrow$ "...hybrid data assimilation (HDA). ..."

*'tandem data assimilation (TDA).'*

18. Eq. (11) is it always a gradient with respect to $x_a$? Should this depend on SFDA or HDA?

*Removed and equations are now only given for the specific experimental setups.*

**3   Referee 2 major comments:**

1. My first major comment is necessary to address, because it will clarify the contributions of this paper. If I understand correctly, the goal of this paper is to estimate the parameters of a dynamical model that will be used for forecasting the states of this dynamical model for long time periods (i.e., on climate timescales). This dynamical model does not have an adjoint, therefore a variational data assimilation approach for estimating these parameters given observations cannot be done. However, a simpler, related dynamical model does have an adjoint, therefore optimization with this adjoint can be used to estimate these parameters, which is does through a process the author's call "synchronization." If this is correct, then this needs to be clarified in the introduction and Section 2. Below are a series of more specific details regarding this comment:

- In the second paragraph of the introduction and first paragraph of Section 2.2, the authors refer to a "cost function," however, having an explicit formula for this cost function, particularly in Section 2, will help

to clarify the author's intention. This will emphasize the need for an adjoint (as well as define the adjoint prior to its definition in Eq.(2)), clearly define the arguments of the cost function for which you intend to minimize, and contextualize this work within the existing variational data assimilation literature. In addition, it would be helpful to clarify whether you are also minimizing such cost function for the state estimate as well, therefore defining a joint state-parameter estimation problem. For example, Chapters 4 and 5 of Evensen et al. (2022), formulate weak constraint and strong constraint 4D-Var data assimilation for the joint state parameter vector z. It would be very helpful to compare what you are doing with standard formulations, such as those presented in this book. With respect to the cost functions defined in 2.4.1-2.4.4, these cost functions look different than the standard 4D-Var cost functions in the data assimilation literature (e.g. like those presented in Desroziers et al., 2014; Evensen et al., 2022). The authors should explain the difference between these cost functions and the cost functions used in 4D-Var, which again will help to clarify the intentions of this work and contextualize it within existing data assimilation literature.

The formulation of the problem follows the strong constrained formulation described in Chapter 4 of Evensen et al. (2022) where the primary goal is to estimate unknown model parameters, and the secondary goal is to improve its state through said parameters. Therefore we are using joint state-parameter estimation problem without a joint vector $z$ as in Evensen et al. (2022). Different from their formulation no background term is employed. Even though prior information is available and it would guarantee a well-posed problem, we consider the limit of infinitely small weights on the background term, such that it does not affect the estimation of the parameters. We don't think this is a problem as we are confident in the observability of the parameters. We added this detail.

- This next comment is regarding the specific details of the experiments: In the first paragraph of Sec. 2.2, the authors use the phrase, "control parameters," however it is unclear if these are are these the model parameters $\sigma, \rho, \beta$ or possible the state variables $x, y, z$. Definition of a cost function in this section would address this question. Second, is this set up correct: the assimilation window is 100 model time units, and over this window only the parameters of the Lorenz '63 are estimated (the state variables $x, y, z$ are not), and this generates estimates of new parameters? What is the frequency of the pseudo-observation time series that is assimilated in this window? After the new parameters are estimated, do the authors perform a forecast of the state with these new parameters to compare with the true model to compute the RMSE? The content of Sections 2 and 3 can be expanded to address these questions, which will help the readers better understand the experiments. This will also help to clarify results presented in figures in Sec. 4.

  References to parameters have been changed to only reference model parameters. Following this comment and others from the first anonymous referee a generic definition synchronisation, the cost function, and the adjoint are introduced in the methodology section. Specific Lorenz '63 details are now only found in the experimental setup section. We also added the following paragraph to section 3.5: 'After parameter estimation, the optimised parameters are used to initialise a free unsynchronised run of the model. The attractors are plotted against the attractor of the true model. In all cases with synchronisation greater than or equal to the optimum value, the attractors' KDE shows precise and consistent agreement with that of the true model. Results are not displayed as there is no differences which are merit discussion. Thus, our focus in the subsequent results is to compare how the examined setups differ in terms of accuracy and precision of optimised parameters recovered. '

- The authors introduce the idea of synchronization: in the abstract, there are facts about synchronization that are described in the abstract (such as reducing positive Lyapunov exponents to negative values) that should also be discussed in Section 2.3, and possibly in the introduction as further motivation for this technique. I suggest adding a more detailed description of synchronization in the beginning of Sec 2.3, particularly after the sentence "The problem can be mitigated by synchronization. . . " Are there any simple examples that can illustrate the synchronization technique one could describe here, before showing how it applies to the Lorenz '63 system?

  We now include a discussion on Lyapunov exponents and have introduced a generic ODE in the methodology to discuss synchronisation, the cost function, and the adjoint. This is found throughout the new form of Section 2.

2. The second major comment I will make is on the literature review and discussion of data assimilation, which begins in the first two paragraphs of the introduction and is discussed in various places throughout the

rest of the manuscript. In order to correctly contextualize and understand the contributions of this work, the authors can expand their literature review on data assimilation. In the second paragraph of the introduction, the authors state that there are two common approaches to data assimilation, "sequential data assimilation" and "variational approach." This is correct, but can improved. Sequential data assimilation should be explained and contrasted with variational data assimilation: if by sequential data assimilation you mean Kalman filters and their variations (e.g, extended Kalman Filter, ensemble Kalman filters, square-root filters), please specify these and cite the appropriate references (for instance, but not limited to Kalman, 1960; Evensen, 1994; Tippett et al., 2003; Houtekamer and Mitchell, 2001) and note that these methods, by design, are adjoint free methods since they do not require minimizing a cost function. With respect to the variational methods, there are other variational schemes in addition to 4D-Var, such as 3D-Var, weak-constraint and strong constraint 4D-Var, and the ensemble-variational filters such as 4DEnVar (Desroziers et al., 2014, and references therein). A brief review of these methods and references should be included.

Paragraphs two and three of the introduction now read: 'There are two common assimilation approaches typically used to incorporate observations into a model: sequential and variational data assimilation schemes (Wunsch, 1996). Sequential data assimilation (Bertino et al., 2003) involves the application of a filter, most commonly Kalman filters (Kalman, 1960; Evensen, 1994, 2003; Tippett and Chang, 2003; Houtekamer and Mitchell, 2001). This technique merges a predicted state with observations at each analysis time step by estimating a joint probability distribution between the two by taking into account their respective modelling and observational uncertainties. Variants of the Kalman filter technique include: extended Kalman filters, ensemble Kalman filters, and square-root filters (Bar-Shalom et al., 2004; Simon, 2006; Evensen, 2003; Van Der Merwe and Wan, 2001; Tippett et al., 2003). They all share a similar basic procedure while differing in case specific variations of the methodology. The strength of all filtering techniques is that the sequential procedure allows for real-time assimilation of observations, for example in initialised numerical weather forecasting.

In contrast, variational data assimilation (Le Dimet and Talagrand, 1986) estimates a joint probability distribution over an extended period of time by minimising a scalar cost function, defined as the quadratic misfit between the model trajectory and all available observations within a defined time window. The most common approaches include four-dimensional variational assimilation (4D-var.) (Rabier and Liu, 2003), three-dimensional variational data assimilation (3D-var.) (Gustafsson et al., 2001), weak and strong constraint 4D-var (Tremolet, 2006; Fisher et al., 2011), and ensemble variational filters including 4DEnVar (Desroziers et al., 2014). Variational data assimilation is a useful technique for solving both initial value and parameter estimation problems (Evensen et al., 2022; Goodliff et al., 2015; Ruiz et al., 2013; Zou et al., 1992). It will be exclusively used in this study.'

3. The use of the word "hybrid data assimilation" particularly in the title and the method introduced in Sec. 2.4.2, either needs to be changed or properly contextualized, as it may be a bit misleading. The term "hybrid data assimilation" exists in the data assimilation literature, first defined in Hamill and Snyder (2000), where the term "hybrid" arose from combining technique from 3D-Var and the EnKF to incorporate flow-dependence in the background estimation error covariance in the 3D-Var update step. To my understanding, the authors here are not performing hybrid data assimilation, but rather introduce a hybrid of their two models for the parameter estimation. To avoid confusion, I request that the authors change this terminology or clearly distinguish what they mean by hybrid from the terminology that already exists in the data assimilation literature.

Changed to 'Tandem data assimilation'.

**4 Referee 2 detailed comments:**

1. In the second sentence of the introduction that beings with "ESMs can be used . . . " Even though ESM is defined in the abstract, it also needs to be redefined in the text and should not start a sentence. This can be rewritten as "Earth system models (ESMs) can be used..."
   'Earth system models (ESMs) can be used to forecast future states of the system'

2. The word "precision" is used throughout (e.g., last sentence of the abstract, fourth paragraph of the introduction, twice on page 4, once on page 11, twice on page 12). I believe the authors mean to use "accuracy" instead, since the authors are looking at errors compared to a truth run. Precision is defined as repeated experiments yielding the same result, though that result may not be accurate.
   We intended to discuss both accuracy and precision as is shown in our plots and will be valuable for real-world applications. The relevant references to precision listed above now include a reference to accuracy.

For example in the introduction we state: 'Therefore, the objective of this paper is to investigate the accuracy and precision of such a synchronised data assimilation approach.'

3. In the first sentence of Sec. 2.1, please cite the original paper Lorenz (1963) in addition to the Yang et al. (2006) paper.

This is now found in Section 3.1: 'In this study, we use the Lorenz '63 system for all our experiments (Lorenz, 1963).'

4. Regarding the pseudo-observations in Sec. 2.2, (1) how often are these observations saved in the time series, and (2) when defining the additive Gaussian noise, can you please specify the mean and variance for the white noise (this may be defined in the last sentence of this paragraph, however this is a bit unclear)?

Section 3.1 now includes: 'This system of equations will be subsequently referred to as the true model with the parameters $\vec{\theta}_t = (10, 28, 8/3)$. This true model is used to generate pseudo-observation which will be used for synchronisation, data assimilation, and parameter estimation. Noise is included in these pseudo-observations by adding random values from a Gaussian distribution centred at zero relative to the true trajectory. The random noise magnitudes are bounded to 25% of the Lorenz '63 system's standard deviation. These pseudo-observations will be labelled as $\vec{x}_o = (x_o, y_o, z_o)$.'

5. In Eq. (2): please define the vector x; I assume this means x = (x, y, z), however this should be defined explicitly either before, along with, or immediately after Eq. (2). The same goes for the vectors x0 and xa in Eq. (7) and xf in Eq. (9).

Section 3.1 now includes: '...where $\vec{x} = (x, y, z)$ are the state variables...'

6. The beginning of Sec. 2.4: what do you mean by "This" in the first sentence? Can you briefly summarize what you mean by this?

We changed the opening to Section 3.3 to: 'A multi-model tandem technique is now considered, which consecutively synchronises two forward models before running the adjoint of the second model backward in time.'

7. Regarding Fig. 1, the figure looks nice, however making the linewidths thicker will make it a bit more readable.

The figure has been rescaled.

8. Section 2.4.2, the sentence ". . . may differ in resolution or numerical formulation by are governed by the same equations" By "same equations" do you mean same continuum dynamics?

Section 3.3.2 now contains a clarification of this point: 'In TDA we assume that both models may differ in resolution or numerical formulation but are governed by the same continuum dynamics.'

9. Regarding Eqs. (8) and (10), it is unclear where these adjoints appear. To improve this, I suggest the authors reorganize section 2.4 by combining sections 2.4.1 with 2.4.4 and 2.4.2 with 2.4.5 so that these equations are adjacent in the text and better explain your methodology to the readers.

This reorganisation was undertaken and the new corresponding subsections can be found in the experimental setup section 3, in the form suggested by this comment.

10. In the second paragraph of the Results, why do the authors plot the 68% percentile? A brief sentence justifying this choice would be useful.

We added the following sentence to Section 3.5: 'The PIs are included to illustrate the statistical spread of the results and reproducibility, not to explicitly indicate uncertainty. Hence, we choose 68% for our PIs to give a concise visualisation of the central $1\sigma$ of results.'

11. In the first second paragraph of Sec. 3.1, the authors say "results of two fits carried out with noise of 25%" can you clarify what it is meant by "noise of 25%" namely what is this noises added to, and what is the 25% being applied to calculate the noise.

Section 4.1 now included the statement: 'Fig. 5 shows the results of two fits carried out for data with an applied noise of 25% relative to the systems' standard deviation.'

12. This is a question that may be of interest: based on your experiments, is there an optimal choice of $\alpha$? For example, Fig. 7 is slightly concave up, indicating a value of $\alpha$ that can minimize the error. This may be worth exploring.

This question has been addressed in the results section 4 as the authors agree that this does suggest an optimal choice for $\alpha$.

13. The first sentence of Sec. 3.3 begins with the acronym "HDA," I suggest that the authors change this sentence so that it does not start with an acronym.
This sentence in Section 4.3 now reads: 'In this section, the tandem data assimilation (setup 2) is be used with different forward and adjoint models that share common physics to examine the impact of introducing model discrepancies.'